



**Scattering and Absorption Cross-sections of Atmospheric Gases in the**
**Ultraviolet-Visible Wavelength Range (307 - 725 nm)**
Quanfu He[1], Zheng Fang[1], Ofir Shoshamin[2], Steven S. Brown[3,4], Yinon Rudich[1,*]
[1] Department of Earth and Planetary Sciences, Weizmann Institute of Science, Rehovot 76100,
Israel
[2] Department of Environmental Physics, Institute for Biological Research, Ness-Ziona 74100,
Israel
[3] Chemical Sciences Division, Earth System Research Laboratory, National Oceanic and
Atmospheric Administration, 325, Broadway, Boulder, CO 80305, USA
[4] Department of Chemistry, University of Colorado, 216 UCB, Boulder, CO 80309, USA
*Correspondence to:* Yinon Rudich (yinon.rudich@weizmann.ac.il)
**Abstract**
Accurate Rayleigh scattering and absorption cross-sections of atmospheric gases are essential for
understanding the propagation of electromagnetic radiation in planetary atmospheres. Accurate
extinction cross-sections are also essential for calibrating high finesse optical cavities and
differential optical absorption spectroscopy and for accurate remote sensing. In this study, we
measured the scattering and absorption cross-sections of carbon dioxide, nitrous oxide, sulfur
hexafluoride, oxygen, and methane in the continuous wavelength range of 307–725 nm using
Broadband Cavity Enhanced Spectroscopy (BBCES). The experimentally derived Rayleigh
scattering cross-sections for $CO_2$, $N_2O$, $SF_6$, $O_2$, and $CH_4$ agree with refractive index-based
calculations, with a difference of 1.5% and 1.1%, 1.5%, 2.9%, and 1.4% on average, respectively.
The $O_2$-$O_2$ collision-induced absorption and absorption by methane are obtained with high
precision at the 0.8 nm resolution of our BBCES instrument in the 307–725 nm wavelength range.
New dispersion relations for $N_2O$, $SF_6$, and $CH_4$ were derived using data in the UV-vis wavelength
range. This study provides improved refractive index dispersion relations, *n*-based Rayleigh
scattering cross-sections, and absorption cross-sections for these gases.





## 1. Introduction

The dominant interactions of gas-phase molecules with light in Earth's atmosphere can be divided into absorption, where the light energy is converted to internal energy and generally (at atmospheric pressures) transferred to the surrounding environment as either as heat or as photoemission, and light scattering where the gases redistribute the light energy in the atmosphere. The knowledge of light extinction (scattering + absorption) by gases is essential for predicting the radiative transfer in the atmospheres of the Earth and other planets. In addition, the light extinction by gases is widely used for determining the effective optical pathlength of high-finesse optical cavities that measure trace gases and aerosols (Washenfelder et al., 2013; Washenfelder et al., 2008; Wilmouth and Sayres, 2019; Jordan et al., 2019) and for Differential Optical Absorption Spectroscopy (DOAS) to infer information about the light extinction properties of aerosols and clouds in the open atmosphere (Baidar et al., 2013; Platt and Stutz, 2008).

The interaction of light with a wavelength much larger than the size of a molecule/particle gives rise to the scattering of light, which is known as Rayleigh scattering(Strutt, 1899). Rayleigh scattering accounts for scattering, local field effects (Lorentz–Lorenz) (Strutt, 1920) as well as depolarization from the non-sphericity of molecule/particles (King correction factor) (King and Eve, 1923; Strutt, 1918). For gas with known refractive index ($n_v$) and King correction factor ($F_k(v)$), the wavelength-dependent Rayleigh scattering cross-section ($\sigma_v$, cm$^2$ molecule$^{-1}$) can be calculated as follows (Sneep and Ubachs, 2005):

$$\sigma_v = \frac{24\pi^3 v^4}{N^2}\left(\frac{n_v^2-1}{n_v^2+2}\right)^2 F_k(v) \tag{1}$$

where $N$ is the number density of the gas (molecules cm$^{-3}$) and $v$ is the wavenumber of the light (cm$^{-1}$). Note that the cross-section contains the gas number density but is not in fact dependent on the number density since the refractive index also appears in the expression. This $n$-based method is an advantageous approach for calculating Rayleigh scattering cross-sections, but it is vital to note that the accuracy of the calculated cross-sections depends on the experimentally-determined refractive index and the King correction factors. In particular, cautions should be used when applying a dispersion formula derived from measurements in one wavelength region to calculate Rayleigh scattering cross-sections in a different wavelength range.





Direct experimental measurement of Rayleigh scattering cross-sections is essential given the
potential uncertainties in *n*-based calculations. While measurements of the King correction factors
and refractive index for gases are well known from the literature (Cuthbertson and Cuthbertson,
1932; Leonard, 1974; Strutt, 1920; Vukovic et al., 1996; Hohm, 1993), there are only a few direct
measurements of Rayleigh scattering cross-sections (Ityaksov et al., 2008a, b; Jordan et al., 2019;
Naus and Ubachs, 2000; Sneep and Ubachs, 2005; Thalman and Volkamer, 2013; Thalman et al.,
2014; Wilmouth and Sayres, 2019; He et al., 2018; Fuchs et al., 2009), especially measurements
with a continuous spectrum from ultraviolet to visible.
Rayleigh scattering cross-section measurements were previously performed at a single wavelength
(e.g., 458 nm, 532 nm, 632.8 nm) using Nephelometry (Shardanand and Rao, 1977) and cavity-
ring down spectroscopy (CRDS) (Ityaksov et al., 2008a, b; Naus and Ubachs, 2000; Sneep and
Ubachs, 2005; He et al., 2018). More recently, advanced Broadband Cavity Enhanced
Spectroscopy (BBCES) was used to determine the Rayleigh scattering cross-sections of gases such
as Ar, $CO_2$, $O_2$, $SF_6$, and $CH_4$. The BBCES technique enables the measurement of Rayleigh
scattering cross-sections over a broad wavelength range. Thalman et al. (2014) performed
measurements over selected wavelength regions between 350 and 660 nm using six BBCES
cavities for $N_2$, Ar, and $O_2$. The BBCES were calibrated with He and $N_2$ using Rayleigh scattering
cross-sections calculated using their refractive index and from cavity-ring down measurements,
respectively. They found a good agreement with *n*-based values to within 0.2±0.4%. Recent studies
using BBCES with 30 nm spectral range were also used for Rayleigh scattering cross-section
measurement in the UV wavelength region and demonstrated excellent agreement with *n*-based
values for Ar and $CO_2$ (Wilmouth and Sayres, 2020, 2019). Recently, Rayleigh scattering cross-
sections for $CO_2$ were measured using BBCES at visible wavelengths between 400 and 650 nm,
and agreement with *n*-based values was within 2.4% on average. To the best of our knowledge,
there is no direct continuous wavelength measurements of extinction cross-sections of gases that
covers the ultraviolet across the entire visible range (300–725 nm) as shown in Table 1.
In this study, we used a recently-developed BBCES instrument to measure the extinction cross-
sections of $CO_2$, $N_2O$, $SF_6$, $O_2$, and $CH_4$ continuously across the wavelength region 307–725 nm.
All of the measurements were done at a single pressure to eliminate effects due to alignment.  This
requires the use of two gases with different Rayleigh cross-sections for the calibration of the
BBCES instrument since the reference state is not vacuum. In this study, He and $N_2$ were used to





calibrate the system. By using the *n*-based calculated Rayleigh scattering cross-sections of He and
$N_2$ to calibrate the path length of the optical cavity, the other cross-sections can be determined
relative to the difference between these two gases. We report high accuracy Rayleigh scattering
cross-sections for all five gases and compared our results with previous *n*-based values. New
dispersion relations for $N_2O$, $SF_6$, and $CH_4$ are derived by incorporating data obtained by this study,
extinction cross-section data in the deep UV, and previously available scattering cross-section data
in the visible wavelength range.

**2. Methods**
**2.1 Extinction measurement using BBCES**
The BBCES systems used in this study are analogous to our previous studies (He et al., 2018;
Washenfelder et al., 2016; Bluvshtein et al., 2016). Briefly, our BBCES consists of two channels,
one in the UV (BBCES$_{UV}$, 307-350 nm) and one in the UV-vis range (BBCES$_{Vis}$, 338-725 nm).
The two channels of the BBCES share a laser-driven Xenon arc lamp source (LDLS EQ−99CAL,
Energetiq Technology, Inc., MA, USA) coupled with a high transmission UV-Vis optical fiber
from which the light is collimated and focused (BBFIBERX-600-1M, Energetiq Technology, Inc.,
MA, USA). The light source was purged with high purity $N_2$ and cooled by an aluminum block
(with 15℃ circulating water inside) to maintain stable optical power output. The UV light from
the fiber is reflected by a low-pass dichroic mirror and filtered into the BBCES$_{UV}$ channel, which
has a cavity with two 2.5 cm diameter, 1 m radius of curvature mirrors, with manufacturer's
reported reflectivity of 0.9995 (per pass loss = 500 parts per million, ppm) at the nominal center
wavelength of 330 nm (Advanced Thin Films, Boulder, USA). The transmitted UV-vis light from
the beam splitter is reflected and filtered into the BBCES$_{Vis}$ channel consisting of two 2.5 cm, 1 m
radius of curvature mirrors (FiveNine Optics, USA) with manufacturer's reported reflectivity
above 0.9993 (loss < 700 ppm), see Figure S1. The light emerging through the rear mirror of the
cavity was focused using a 0.1 cm F/2 fiber collimator (74−UV, Ocean Optics, Dunedin, FL, USA)
into a high transmission UV-vis optical fiber which directs the light into a high-performance
spectrometer (QEPro, Ocean Insight, USA). Before gas measurement, the wavelength of the
spectrometer was calibrated using an HG-1 mercury argon calibration light source (Ocean Insight,
USA) within the wavelength range of 302.15−727.29 nm. During these experiments, a 300 line



mm$^{-1}$ grating and a 200 μm entrance slit width were used. The CCD array is a back-illuminated
detector with 1024×56 pixels (Hamamatsu S7031-1006, Japan) thermo-electrically cooled to −10 ℃
to reduce thermal noise. Individual spectra at a wavelength resolution of 0.8 nm were acquired
with 3.0 s integration time, and a total of 150 spectra were recorded during each measurement.
During the extinction measurements, the entire 94.0 ± 0.1 cm pathlength between the mirrors was
filled with He, $N_2$, $CO_2$, $N_2O$, $SF_6$, or $CH_4$. The gases were obtained from several vendors (Airgas,
Linde) with the following purities: He, 99.995%; $N_2$, 99.999%; $N_2O$, 99.999%, $CO_2$, 99.999%;
$SF_6$, 99.999%; $CH_4$, 99.9995%.
The reflectivity of the mirrors (R($\lambda$)) can be determined as a function of wavelength ($\lambda$) by taking
into account the difference in the extinction due to known literary data of Rayleigh scattering
coefficient ($\alpha_{Ray}^{gas}$) by two different gases such as $N_2$ ($\alpha_{Ray}^{N_2}(\lambda)$) and He ($\alpha_{Ray}^{He}(\lambda)$) (Washenfelder
et al., 2008).
$$\frac{1-R(\lambda)}{d} = \frac{I_{N_2}(\lambda)\left(\alpha_{Ray}^{N_2}(\lambda)\right) - I_{He}(\lambda)\left(\alpha_{Ray}^{He}(\lambda)\right)}{I_{He}(\lambda) - I_{N_2}(\lambda)} \quad (2)$$
where $d$ is the length of the cavity filled by the gas. In this study, the studied gas filled the entire
length of the cavity (94.0 ± 0.1 cm) since no purge flows were used. $I_{gas}$ is the light intensity
measured by filling the cavity with high purity $N_2$ ($I_{N_2}(\lambda)$) and He ($I_{He}(\lambda)$). Rayleigh scattering
($\alpha_{Ray}^{gas}$) is the combined product of Rayleigh scattering cross-section ($\sigma$) and the gas number density
($N$) during the measurements. Rayleigh scattering cross-sections of $N_2$ and He are calculated using
the data in Table 1. Figure S1 shows typical examples of light intensity when the BBCES cavities
are filled with pure $N_2$. Reflectivity measurements were repeated every three sample measurements
to track the stability of the system.
Once the reflectivity is determined, it is possible to calculate the wavelength-dependent extinction
cross-sections of other gases ($\sigma(\lambda)$) as follows:
$$\sigma(\lambda) = \left[\left(\frac{1-R(\lambda)}{d}\right)\left(\frac{I_{He}(\lambda)-I_{gas}(\lambda)}{I_{gas}(\lambda)}\right) + \left(\frac{I_{He}(\lambda)}{I_{gas}(\lambda)}\right)\left(\alpha_{Ray}^{He}(\lambda)\right)\right]/N \quad (3)$$
Where $N$ is the number density of the gas during the measurements, and $I_{gas}(\lambda)$ is the light intensity
when a target gas fills the cavity. During our experiments, the purge flow of the high reflection





mirrors was shut down to ensure that the cavity was filled with target gas completely. To measure
the extinction cross-sections of $CO_2$, $N_2O$, and $SF_6$, the cavity is filled with pure target gas. Mass
flow controller controlled $O_2$/$CH_4$ flow was mixed with He in a 2 m Teflon tube ($\Phi$ = ¼ inch) to
generate a gas mixture with total flow rate of 500 mL min$^{-1}$. For $O_2$ experiments, measurements
were performed for $O_2$ + He mixtures by varying the $O_2$ percentage between 10% and 100% with
a 10% step. The $CH_4$, measurements were performed for $CH_4$ + He mixtures with $CH_4$ percentage
ranges between 10% to 100% with a 10% step. Additional measurements were also performed for
15%, 25%, 35%, and 45% $CH_4$.

**2.2 Extinction measurements using cavity-ring down systems (CRDS) at 404 nm and 662 nm.**

To obtain independent measurements for the extinction cross-sections and to cross-validation of
the BBCES technique, we conducted CRD measurements at two fixed wavelengths of 404 nm and
662 nm. CRDS is a highly sensitive technique and uses a different measurement principle than
BBCES. The CRDS measured the decay rate of light due to extinction rather than an absolute
absorbance (as in the BBCES) and thus immune to shot-to-shot source light fluctuations. A
detailed description of the CRD method for light extinction measurement can be found in
Bluvshtein et al. (2016) and He et al. (2018). Briefly, diode lasers (110 mW 404 nm diode laser,
iPulse, Toptica Photonics, Munich, Germany; 120 mW 662 nm diode laser, HL6545MG, Thorlabs
Inc., NJ, USA) are used as the light source of these CRDS. The 404 nm and 662 nm lasers are
modulated at 1383 Hz and 500 Hz with a 50% duty cycle. The diode lasers are optically isolated
by quarter waveplates (1/4 λ) and polarizing beam splitters to prevent damage to the laser head by
back reflections from the highly reflective CRDS mirror. The back-reflected light beam is directed
into a photodiode, which serves as an external trigger source. Light transmitted through the back
mirror of the cavity is collected by an optical fiber and detected by a photomultiplier tube (PMT),
which samples at a rate of 10 to 100 MHz. The time-dependent intensity data is acquired with a
100MHz card (PCI-5122, National Instruments, USA) and processed by data acquisition software
in Labview. An exponential curve is fitted to each intensity decay data set (Figure S2). Over 1000
decay time measurements are monitored and averaged on a second basis. The residual of the fit
for the averaged intensity decay is obtained and further normalized to the averaged intensity. The
derived relative residuals (Figure S2) show no apparent structure with other time constants,
validating the application of CRDS as a good measure of extinction. The resultant 1 Hz decay time



is averaged over one measurement duration of five minutes with standard error as the measurement
uncertainty.
All of the CRDS measurements were performed under room temperature and pressure downstream
from the BBCES instrument. The gas temperature (K-type thermocouple) and cavity pressure
(Precision Pressure Transducer, Honeywell International Inc., MN, USA) were recorded for gas
number density ($N$) calculation. During the CRDS measurements, the full cavity was filled with
the investigated gases (He, $CO_2$, $N_2O$, $SF_6$, $O_2$, $CH_4$, or gas mixtures ($O_2$ + He and $CH_4$ + He)).
The extinction cross-section ($\sigma(\lambda)$) of the studied gas was measured relative to that of He and was
calculated by  equation (4):
$$\sigma(\lambda) = \frac{L}{clN}\left(\frac{1}{\tau_{gas}} - \frac{1}{\tau_{He}}\right) + \sigma_{He} \tag{4}$$
Where $L$ is the total length of the cavity ($l$), $c$ is the speed of light, and $\tau_{gas\,and\,\tau_{He}}$ are the ring-down
time of the cavity when it is filled by target gas or by the reference gas, He.

**2.3 Data processing**

For comparison, the scattering cross-sections of the gases investigated in this study were also
calculated with Equation (1) based on the refractive index and the King correction factors available
in the literature that are listed in Table 1. The King correction factors are taken as unity for mono-
atomic molecules and spherical molecules (with regards to the depolarization) but deviates for
non-spherical molecules. For the 307–725 nm wavelength range of this study, the $n$-based
calculated Rayleigh scattering cross-sections from largest to smallest are $SF_6$, $N_2O$, $CO_2$, $CH_4$, $N_2$,
$O_2$, and He.
The extinction of $O_2$ + He mixtures ($\alpha_{O_2+He}$) consists of the extinction by $O_2$ ($\alpha_{O_2}$) and He ($\alpha_{He}$),
and the $O_2$–$O_2$ collision-induced absorption ($\alpha_{O_2-O_2}$). The extinction of $O_2$ and He is a combined
product of extinction cross-section ($\sigma_{gas}$) and gas number density ($N_{gas}$). Thus $\alpha_{O_2+He}$ can be
described with the following equation:
$$\alpha_{O_2+He} = \sigma_{O_2-O_2} \times N_{O_2}^2 + \sigma_{O_2} \times N_{O_2} + \sigma_{He} \times N_{He} \tag{5}$$
Where $N_{O_2}$ and $N_{He}$ are the number density of the $O_2$ and He in the cavities. Performing a $2^{rd}$ order
polynomial fit to the extinction obtained by the BBCES with respect to the gas number density





thus yields the extinction cross-section of $O_2$ and the $O_2$-$O_2$ collision-induced absorption (CIA)
cross-section.
In addition to the results from $2^{rd}$ order polynomial fitting, we also used data from pure $O_2$
measurement to calculate the extinction by $O_2$ and by CIA of $O_2$–$O_2$. The real refractive index of
$O_2$ ($n_{O_2}$) derived from extinction data measured in the wavelength regions where there is no
absorption was fitted using the generalized expression of $(n_{O_2} - 1) \times 10^8 = A + \frac{B}{C - \nu^2}$. Based on
the refractive index, the scattering cross-sections of $O_2$ in the wavelength range of 307-725 nm
were further calculated. By subtracting the scattering cross-section of $O_2$ from the measured total
extinction, we derived the CIA of $O_2$–$O_2$. However, the $O_2$ absorption bands at 580, 630, and 690
nm overlaps with those of $O_2$–$O_2$ collisions. Additional corrections are thus needed to split the
absorption by $O_2$ and $O_2$–$O_2$ collision, which is out of the scope of this study.
Methane has weak vibrational overtone absorption in the UV-vis wavelength range that is
comparable to or greater than its Rayleigh scattering. Previous high-resolution spectroscopy
studies have identified smooth and unstructured absorption bands across the UV-visible range
(Giver, 1978; Smith et al., 1990). The spectral features are substantially broader than 0.8 nm, thus
the absorption by $CH_4$ can be measured by our BBCES. The measured extinction coefficients of
$CH_4$+He mixtures ($\alpha_{CH_4+He}$) are linearly correlated with the number concentration of the $CH_4$ ($N_{CH_4}$)
as described by the following equation:
$$\alpha_{CH_4+He} = \sigma_{CH_4} \times N_{CH_4} + \sigma_{He} \times N_{He} \qquad (6)$$
A linear fit was used for deriving the extinction cross-section of $CH_4$. The absorption between 300
and 400 nm is negligible as compared to the Rayleigh scattering. Thus extinction data in this UV
wavelength range were used to calculate the real part of the refractive index of $CH_4$ which was
further fitted utilizing the expression of $(n_{CH_4} - 1) \times 10^8 = A + \frac{B}{C - \nu^2}$. By applying this
dispersion relation, the Rayleigh scattering cross-sections in the entire wavelength range of 307–
725 nm were derived. Finally, the $CH_4$ absorption cross-sections were calculated by subtraction of
the scattering cross-section from the extinction cross-section.
**2.4 Error Propagation for BBCES**





The uncertainty for BBCES measurements can be assessed by the propagation of the errors
associated with the measurements. The pressure (±0.01%), temperature (±0.1%) and cavity length
(94.0 ± 0.1 cm) are combined with the Rayleigh cross-section uncertainties for $N_2$ (±1%) as well
as uncertainty in the measurements of the spectral signal by the spectrometer (≪0.2%) to get an
overall relative uncertainty for the mirror reflectivity curve of ±1.03%. This uncertainty is further
propagated to the target gas by consideration of the uncertainties of pressure, temperature, and
spectral intensity of the target gas measurements. The overall 1-σ uncertainty of the gas extinction
cross-section is 1.1%. The precision of the mass flow controllers is 0.5 mL min$^{-1}$. When the total
flow rate is 500 mL min$^{-1}$, the resulted uncertainty of the gas concentration (10-100%) varies from
0% to 1.0%. Thus, the overall 1-σ uncertainty of extinction coefficients measured for $CH_4$+He and
$O_2$+He varies from 1.1% to 1.5%. The detailed wavelength-dependent uncertainties were
calculated due to the wavelength-dependence of the spectral intensity. The results are shown and
discussed in later sections.

**3 Results and Discussion**
**3. 1 Performance of the optical system**
The reflectivity of the cavity mirrors, measured across the entire range using the difference in
Rayleigh scattering of $N_2$ and He, was very stable throughout the experiments. The measured
mirrors reflectivity curves are shown in Figure S1. The mean peak reflectivity of the $BBCES_{UV}$
mirrors was 0.999328±0.000006 (672±6 ppm) at 330 nm, with a corresponding effective optical
pathlength of 1.40±0.01 km. The reflectivity curve of the $BBCES_{Vis}$ is much more structured, with
reflectivity ranging between 0.999224±0.000010 and 0.9999550±0.0000006 (776±10 ppm > loss >
45±0.6 ppm) over a wide wavelength range of 338–725 nm. The reflectivity of the $BBCES_{Vis}$ is
much higher than that of our previous system (He et al., 2018) and also covers a much broader
wavelength range. Thus the effective pathlength of the BBCESVis varies between 1.3 and 20.4
km, guaranteeing a high sensitivity of the extinction measurement. The mean uncertainty in the
effective pathlength across the measured wavelengths as determined from the mirror reflectivity
was ±1.03%, which is predominantly due to the uncertainty in the Rayleigh scattering cross section
for $N_2$ derived from *n*-based calculation.
**3. 2 Rayleigh scattering cross-sections of $CO_2$, $N_2O$, $SF_6$.**



Figure 1 shows the extinction cross-sections of $CO_2$, $N_2O$, and $SF_6$ measured by the BBCES. The
extinction cross-sections of these gases monotonically decrease with increasing wavelength, and
no absorption (i.e., no structured extinction larger than the smoothly varying Rayleigh curve) is
observed in the wavelength range of 307–725 nm, indicating that the measured extinction is due
solely to the Rayleigh scattering of these gases. The wavelength-dependent relative standard
deviations of the measurements for each gas are shown in Figure 1d. The mean 1-σ uncertainty of
the reported cross sections for all three gases across the 307–725 nm wavelength range is 1.5% for
$CO_2$, 1.1% for $N_2O$, and 1.5% for $SF_6$. As mentioned above, the derived uncertainty originates
predominantly from the uncertainty in the $N_2$ Rayleigh scattering cross-section. Uncertainty in the
Rayleigh cross-sections of each gas varies with wavelength and generally tracks the light intensity
spectra, which is a combined product of light source spectrum and the mirror reflectivity profile.
The uncertainty is much higher when the transmitted light intensity is low (Figure S1).
The BBCES measured Rayleigh scattering cross-sections for these three gases agree well with
those obtained by our CRDS operating at 404 nm and 662 nm, with deviations smaller than 1.6%.
Table 2 listed the Rayleigh scattering cross-sections at several wavelengths obtained by the
BBCES measurements (Exp) and by the theoretical calculations using the refractive index and $F_k(\nu)$
values from Table 1 (*n*-based). The relative differences between these two sets of results are within

274 1.4%.

Figure 1a–c shows a comparison of the measured Rayleigh scattering cross-sections for $CO_2$, $N_2O$,
and $SF_6$ with *n*-based calculations and with previous experimental results from the literature. There
are a few measurements for the Rayleigh scattering cross-sections for $CO_2$ which cover a wide
spectral range (Jordan et al., 2019; Shardanand and Rao, 1977; Sneep and Ubachs, 2005; Wilmouth
and Sayres, 2019; He et al., 2018). There are fewer Rayleigh scattering measurements for $N_2O$ and
$SF_6$ in the studied wavelength range. The measured Rayleigh scattering cross-sections for $CO_2$,
$N_2O$, and $SF_6$ are in excellent agreement with *n*-based calculation. The wavelength-dependent
difference of our experimentally derived Rayleigh scattering cross-sections with *n*-based
calculations are shown in Figure 1e. The mean ratios of our measurements to the n-based values
for the entire wavelength range of 307–725 nm are 1.00 ±0.01, 0.99±0.01, and 1.01±0.01 for $CO_2$,
$N_2O$, and $SF_6$, respectively. Notably, while our results for $N_2O$ agree well with the *n*-based
calculations, previous results obtained by CRDS at 532 nm (Sneep and Ubachs, 2005) and by
absorption spectroscopy in the wavelength of 300–315 nm (Bates and Hays, 1967) do not agree





well with the n-based calculations. The measurements between 300 and 315 nm were first
published by Bates and Hays (1967), who obtained the results from a doctoral thesis. However,
the results from our BBCES system are in good agreement with the *n*-based calculations and with
experimental results from independent CRDS measurements, thus increasing the confidence in our
measured values.
**3. 3 Scattering and absorption cross-sections of $O_2$.**
The UV-vis spectra of gas-phase molecular oxygen are characterized by discrete structured
absorption bands due to the electronic transition $(b^1 \sum_g^+ (v' = 1/2/3) \leftarrow \sum_g^- (v'' = 0))$ of $O_2$
monomer, broader unstructured CIA of $O_2$–$O_2$, and structured dimer bands from the bound van
der Waals $O_2$ dimer (Newnham and Ballard, 1998). Under atmospheric conditions, the $O_2$–$O_2$ CIA
bands are frequently described as "$O_4$" bands, although absorption by $O_2$ dimer is thought to be
significant only under very low-temperature conditions (Thalman and Volkamer, 2013; Long and
Ewing, 1973). Within the wavelength range investigated in this work, the molecular oxygen B
band at 688 nm $(b^1 \sum_g^+ (v' = 1) \leftarrow X^3 \sum_g^- (v'' = 0))$, γ overtone band at 629 nm $(b^1 \sum_g^+ (v' = 2) \leftarrow$
$X^3 \sum_g^- (v'' = 0))$, and δ overtone band at 580 nm $(b^1 \sum_g^+ (v' = 3) \leftarrow X^3 \sum_g^- (v'' = 0)$ overlap with
$O_2$–$O_2$ CIA bands of $^1\sum_g^+ (v = 1)$, $^1\Delta_g + ^1\Delta_g (v = 0)$, and $^1\Delta_g + ^1\Delta_g (v = 1)$, respectively.
These absorption bands can only be resolved by a high-resolution spectroscopic technique.
Absorption cross-sections of the B, γ, and δ bands were convoluted from the HITRAN database
(Gordon et al., 2017) by considering the temperature, pressure, and instrument's wavelength
resolution. The wings of the oxygen lines also show a quadratic dependence on the pressure due
to pressure broadening. However, due to the minimal $O_2$ absorption contribution below 680 nm
and the low instrument wavelength resolution, the extinction cross-section of the $O_2$ monomer can
be treated as linearly correlated with the $O_2$ concentration. Moreover, the $O_2$–$O_2$ CIA cross-section
is correlated with the square of the $O_2$ concentration. Therefore, these cross-sections can be
retrieved from measurements at different $O_2$ concentrations. Due to the discrete structured
absorption bands and the instrument's wavelength resolution, the range of absorption cross-
sections spans several orders of magnitude within the spectral response of the instrument, limiting
the relevance of the absorption cross-sections for other researchers. These results are not further
discussed here. However, the data for broader unstructured CIA of $O_2$–$O_2$ are still useful for
various applications.



318 Figure 2 shows the wavelength-dependent extinction coefficients of $O_2$+He mixtures. He was used
319 in these experiments to minimize extinction contributions from Rayleigh scattering. Nine
320 absorption peaks centered at 344 nm (CIA), 360 nm (CIA), 380 nm (CIA), 446 nm (CIA), 477 nm
321 (CIA), 532 nm (CIA), 577 nm ($\delta$ overtone and CIA), 629 nm ($\gamma$ overtone and CIA), and 688 nm
322 (B band and CIA) were observed in the wavelength range of 307–725 nm. The absorption
323 coefficients of the central wavelengths for the first eight peaks increase non-linearly with $O_2$
324 concentration while that of the 688 nm peak increases in a more linear manner, indicating that the
325 $O_2$ B band absorption dominates the last absorption peak while the other peaks are mostly
326 associated with CIA of $O_2$-$O_2$.

327 The extinction coefficients obtained by the BBCES correlated well with those measured by the
328 CRD, with slops of 0.990 ($R^2$=0.9994) and 0.993 ($R^2$ = 0.9996) at the wavelengths of 404 nm and
329 662 nm, respectively (Figure 3). This excellent agreement between the instruments further
330 substantiates the BBCES measurements and suggests that the accuracy of the BBCES is better
331 than estimated in the error propagation above, where the $N_2$ refractive index was the largest
332 uncertainty. As explained in the data processing section, the measured extinction coefficients were
333 fitted with a $2^{rd}$ order polynomial (selected wavelengths at the peaks of the CIA absorption bands
334 are shown in Figure 4). At 476.7, 577.2, and 629.2 nm, the absorption is from the CIA of $O_2$–$O_2$.
335 The fit generates positive values matching the absorption cross-section of $O_2$–$O_2$ CIA. At 687.7
336 nm where strong B-band absorption appears, the fit yields a small negative coefficient for $O_2$–$O_2$
337 CIA.

338 Figure 5a shows the extinction cross-section measured for 100% $O_2$. These results agree well with
339 previously reported results by Jordan et al. (2019). For wavelengths where no absorption is
340 detected, the measured extinction cross-sections agree well with $n$-based calculations. Figure 5b-
341 c shows the determined extinction cross-sections for molecular $O_2$ and the absorption cross-
342 sections of $O_2$–$O_2$ CIA. For wavelength ranges without $O_2$ bands, our extinction cross-sections
343 agree well with $n$-based values with an average deviation of 2.8%. The absorption cross-sections
344 for $O_2$–$O_2$ CIA derived in this study mostly agree well with literature data from Thalman and
345 Volkamer (2013). The differences are within 1.1% at 477, 532, 577, and 630 nm but larger
346 deviations were found at 344 (4.2%), 360 (-29%), 380 (-21%), and 446 (4.2%) nm. These
347 absorption bands are the lowest intensity bands and therefore have the largest relative uncertainties
348 in either measurement.





The Rayleigh scattering cross-sections of molecular $O_2$ derived from the 100% $O_2$ measurement
agree well with *n*-based calculations with an average difference of 1.2%. CIA of $O_2$–$O_2$ calculated
from this single measurement matches the results from the fitting method. Due to strong absorption
from $O_2$ B band and $\gamma$ overtone band, this method cannot derive the cross-sections of CIA of $O_2$-
$O_2$ at 630 and 688nm.

### 3. 4 The scattering and absorption cross-sections of $CH_4$.

$CH_4$ has weak absorption in the UV–vis wavelength range, and these bands dominate the
photographic spectra of planets such as Uranus and Neptune. Figure 6 presents the wavelength-
dependent extinction coefficients of $CH_4$+He mixtures. A total of eleven absorption bands were
detected in the wavelength range of 307–725 nm. The extinction coefficients increase as a function
of increasing $CH_4$ concentration. Extinction coefficients obtained by the BBCES correlated well
with those measured in parallel by the CRDS, with slopes of 1.002 ($R^2$=0.9999) and 0.99 ($R^2$ =
0.999) at the wavelengths of 404 nm and 662 nm (Figure S3). The excellent agreement between
these three systems further supports the accuracy of BBCES extinction measurements over a wide
working range. The measured extinction coefficients were linearly fit against the $CH_4$ number
concentration. Figure 7 shows the fitted curves at five selected wavelengths. The extinction
coefficients have a linear correlation with $CH_4$ concentration ($R^2 > 0.9988$) without exception. The
calculated slopes represent the extinction cross-sections of $CH_4$ and also indicate a wide dynamic
range of our BBCES.
The extinction cross-sections for $CH_4$ retrieved from concentration-dependent measurements are
plotted in Figure 8a. BBCES results from this study agree well with results from previous studies
using BBCES (Jordan et al., 2019; Wilmouth and Sayres, 2019) and CRDS (Sneep and Ubachs,
2005). Previous studies using a Nephelometer (Shardanand and Rao, 1977) and interferometer
(Cuthbertson and Cuthbertson, 1920; Watson et al., 1936) obtained the scattering cross-sections
of $CH_4$. The BBCES measures the extinction cross-section. For wavelengths where extinction is
dominated by Rayleigh scattering ($< 475$ nm), our BBCES results agree well with the results from
Nephelometer and interferometer measurements. In this study, the refractive index of $CH_4$ was
calculated using the extinction data in the wavelength range of 307-400 nm. The calculated
refractive index was fitted to the general expression:
$$(n_{CH_4} - 1) \times 10^8 = 5476 + \frac{4.1579\times10^{14}}{1.1568\times10^{10}-\nu^2}$$   (7)
As shown in Figure 8b, our calculated scattering cross-sections are in good agreement with those
derived from the newest refractive index developed by Wilmouth and Sayres (2020) (Table 2),
with an average difference of 1.4%. The absorption cross-section, which is the difference between
the total extinction and the Rayleigh scattering cross-section, is shown in Figure 8c. At most
spectral ranges, our results are in good agreement with the results from previous studies (Giver,
1978; Smith et al., 1990). For example, the difference as compared to the results from Giver (1978)
at 542, 576.4, 598, 619, 665.7, and 703.6 nm is 4.0% on average. At several wavelength regions
(e.g., 520–536nm, 580–605 nm), the results from Fink et al. (1977) differ from all of the other
studies. In the wavelength range of 400–725 nm, absorption contributes up to 99.7% of the $CH_4$
extinction.
**3. 5 Dispersion relations for $N_2O$, $SF_6$ and $CH_4$.**
**$SF_6$:** Wilmouth and Sayres (2020) found that their measured Rayleigh scattering cross-sections for
$SF_6$ in the ultraviolet range were much lower than those from the *n*-based expression of Sneep and
Ubachs (2005). To better constrain the dispersion formula when extrapolated over a broad
wavelength range, we employed an alternative fit of the form $A+B/(C–v^2)$ to our data. The
Rayleigh scattering derived refractive index in the wavelength range of 264-297 nm and 333-363
nm by Wilmouth and Sayres (2020, 2019), and direct refractive index measurement at 632.99 nm
(Vukovic et al., 1996) were used (Figure 9a). The resulting dispersion relation for $SF_6$ in the
wavelength range of 264–725 nm is
$$(n_{SF_6} - 1) \times 10^8 = 22871 + \frac{8.0021 \times 10^{14}}{1.6196 \times 10^{10} - v^2} \tag{8}$$
**$N_2O$:** Sneep and Ubachs (2005) derived the refractive index based on polarizability measurements
using interferometer at five single wavelengths (457.9, 488, 514.5, 568.2, 647.1 nm) by Alms et
al. (1975). In this study, we calculated the refractive index of $N_2O$ from the Rayleigh scattering
cross-sections in the wavelength range of 307–725. Based on this refractive index data set, the
dispersion relation (Eq (9)) for $N_2O$ was retrieved for a much broader wavelength range (Figure
9b) compared to that generated by Sneep and Ubachs (2005).
$$(n_{N_2O} - 1) \times 10^8 = 23154 + \frac{1.534 \times 10^{14}}{6.5069 \times 10^9 - v^2} \tag{9}$$
**$CH_4$:** The previous study by Wilmouth and Sayres (2019) has shown that their measured Rayleigh
scattering cross-sections for $CH_4$ are in substantial disagreement (22%) with those calculated from



the refractive index recommended by Sneep and Ubachs (2005). Sneep and Ubachs (2005)
formulated the refractive index of $CH_4$ based on interferometric measurements at wavelengths of
325, 543.5, 594.1, 612, and 633 nm by Hohm (Hohm, 1993). However, the Rayleigh scattering
cross-sections calculated from their refractive index are much higher than all the measured values
listed in Figure 9b. Using Rayleigh scattering cross-sections in the wavelength range of 264–297
nm, 333–363 nm (Wilmouth and Sayres, 2019, 2020), 307–400 nm from this study, and single
wavelength measurements which are not impacted by absorption (Cuthbertson and Cuthbertson,
1920; Shardanand and Rao, 1977; Watson et al., 1936), we derived the dispersion formula for the
refractive index of $CH_4$ in the combined UV/visible range (Figure 9c) as follows:
$$(n_{CH_4} - 1) \times 10^8 = 7327.7 + \frac{4.1884 \times 10^{14}}{1.2208 \times 10^{10} - v^2} \tag{10}$$
The calculated Rayleigh scattering cross-sections using the dispersion relations derived in this
study were compared with those derived from previously recommended formulations listed in
Table 1 (Figure 9). The difference increases significantly at the longer wavelength in the region of
320–725 nm (Figure S4). The average deviations are 0.8%, 0.9%, and 1.6% for $SF_6$, $N_2O$, and $CH_4$,
respectively. Notably, the difference for $CH_4$ is much more significant than for the other two gases.
This study uses additional measurements and literature data in the wavelength ranges of 307-
333nm and 363–400 nm than those used by Wilmouth and Sayres (2020). Therefore this fit
captures well our BBCES measurements (Figure 9d), and also the Wilmouth and Sayres (2020,
2019) data.
**Conclusions and Implications**
Rayleigh scattering cross-sections between 307 and 725 nm were determined for $CO_2$, $N_2O$,
$SF_6$, $O_2$, and $CH_4$ by simultaneous BBCES and CRDS measurements. Extinction coefficients
obtained by the BBCES show high consistency with those measured by parallel CRDS at 404 and
662 nm (Figure 3 and figure S3), demonstrating that the BBCES measurements provide results
with both a wide wavelength range and high accuracy. Comparison of our measurements with *n*-
based calculations for these gases in the entire wavelength range of this study yields excellent
agreement with relative differences of 1.5% and 1.1%, 1.5%, 2.9%, and 1.4% on average,
respectively. The $O_2$-$O_2$ CIA cross-sections obtained from the BBCES measurements are
compared with those published by Thalman and Volkamer (2013). The relative differences are
within 1.1% at 477, 532, 577, 630 nm. Larger relative differences occur at the weak bands at 344



(4.2%), 360 (–29%), 380 (–21%), and 446 (4.2%) nm. The absorption cross-sections of $CH_4$ in the
wavelength range of 400-725 nm agree well with those documented by Giver (1978).

440       Rayleigh scattering cross-sections of $CO_2$ determined using BBCES in this study, and in other

studies have shown that the refractive index recommended by Sneep and Ubachs (2005) is suitable
for use in the wavelength range of 307–725 nm. By incorporating the refractive index data from
previous studies, we developed new dispersion relations for the refractive index of $N_2O$ (307-725
nm), $SF_6$ (264–725 nm), and $CH_4$ (264–671 nm). The new dispersion relation for $CH_4$ captures the
measurements from BBCES more adequately.

446       Previous studies measured the Rayleigh scattering and absorption cross-sections of $CO_2$, $N_2O$,

$O_2$, $SF_6$, and $CH_4$ at narrow spectral ranges or single wavelengths. In this study, we used BBCES
that covers the broad wavelength range of 307–725 nm to measure total extinction (the sum of
absorption and scattering). The measurements validate that refractive index-based methods for
calculating Rayleigh extinction cross-sections are accurate and provide new fits over more
continuous and extended wavelengths range than available in the literature to constrain such
methods. The Rayleigh scattering cross-sections reported here are useful in several applications.
These include calibration standards based on extinction for optically-based instruments, such as
those designed for aerosol optical properties measurements or trace gas concentrations in the field
(Jordan et al., 2019; Min et al., 2016; Bluvshtein et al., 2017), especially when high-refractive
index gases are used for improved calibration. They will also improve the accuracy of Rayleigh
scattering parameterizations for major greenhouse gases in Earth's atmosphere, $CO_2$, $CH_4$, and
$N_2O$. Accurate quantitative measurements of Rayleigh scattering coefficients and absorption
cross-sections of atmospheric gases such as molecular $N_2$, $O_2$, $CO_2$ and the CIA of $O_2$–$O_2$ cross-
sections in the UV-NIR range are of particular importance for the application of Rayleigh LIDAR
systems, especially at the Nd:YAG laser harmonics 1064, 532 & 366 nm. These systems analyze
the molecular backscattering contributions to the LIDAR's attenuated backscatter signals to
retrieve the atmospheric profile of aerosols and clouds in the planetary boundary layer (Tomasi et
al., 2005; Herron, 2007). Recent NASA satellite missions have also aimed to measure global
carbon dioxide concentrations with high precision (0.25%) (Drouin et al., 2017). These $CO_2$ global
missions use the $O_2$–$O_2$ CIA underneath the structured $O_2$ A-band (760 nm) to evaluate the solar
radiation double pathlength in the Earth atmosphere and to determine the atmospheric pressure.
The measurements in this study validate the existing literature on the extinction of $O_2$ collision





complexes and molecular oxygen bands, and can be used for calibration purposes in both remote
sensing and *in-situ* spectroscopic applications in the atmosphere. In the future, gas extinction
measurements at extended wavelengths (near-infrared) and for additional gases (e.g., $N_2$) will
improve the spectroscopic applications in atmospheric studies.

**Data availability.**

Data are available upon request from the corresponding author (yinon.rudich@weizmann.ac.il).

**Author contributions.**

Q.H., S.S., and Y.R. designed this study. Q.H., Z.F., and O.S. conducted the experiments. Q.H.
prepared the draft and all of the co-authors reviewed it and provided comments.

**Competing interests.**

The authors declare that they have no conflict of interest.

**Acknowledgments**

This research was partially supported by the US-Israel Binational Science Foundation (BSF grant
#2016093). Dr. Q. H. is supported by the Koshland Foundation and the Center for Planetary
Sciences, Weizmann Institute of Science. Dr. Z.F. is supported by SAERI initiative of the
Weizmann Institute.





Figure 1. Rayleigh scattering cross-sections of $CO_2$ (a), $SF_6$ (b), and $N_2O$ (c). Panel (d) shows the
relative standard deviations as a function of wavelength for each gas. The relative difference in the
cross-sections obtained by the BBCES and calculations from the refractive index are displayed (e).

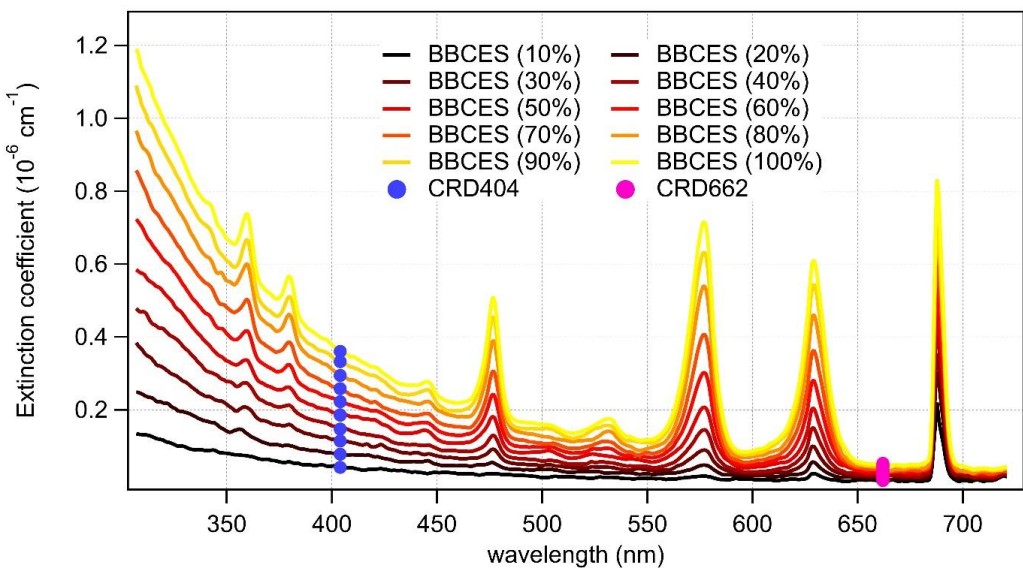


Figure 2. Wavelength-dependent extinction coefficients of $O_2$ + He mixtures as a function of $O_2$
concentration. The colored lines represent the extinction coefficients measured by BBCES, and
markers represent results from CRDS.



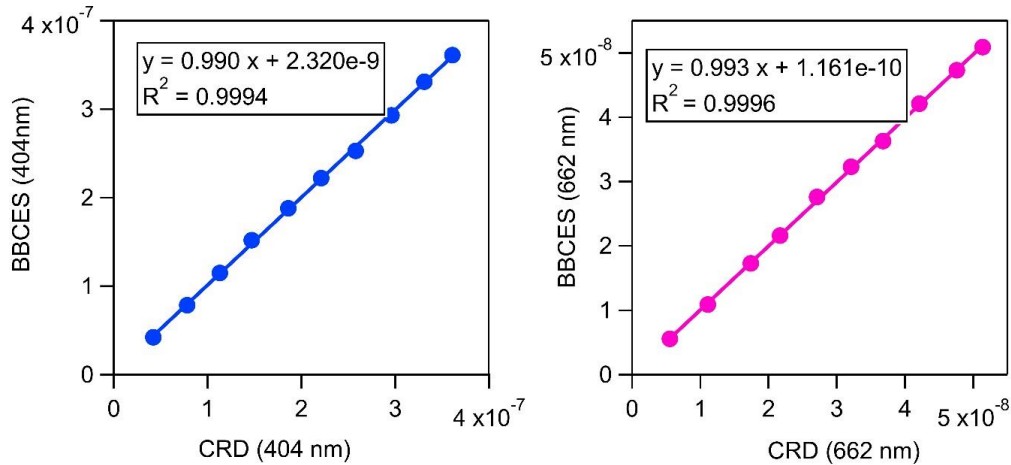


Figure 3. Correlations between the extinction coefficients (unit, $cm^{-1}$) measured by the BBCES
and CRDS.





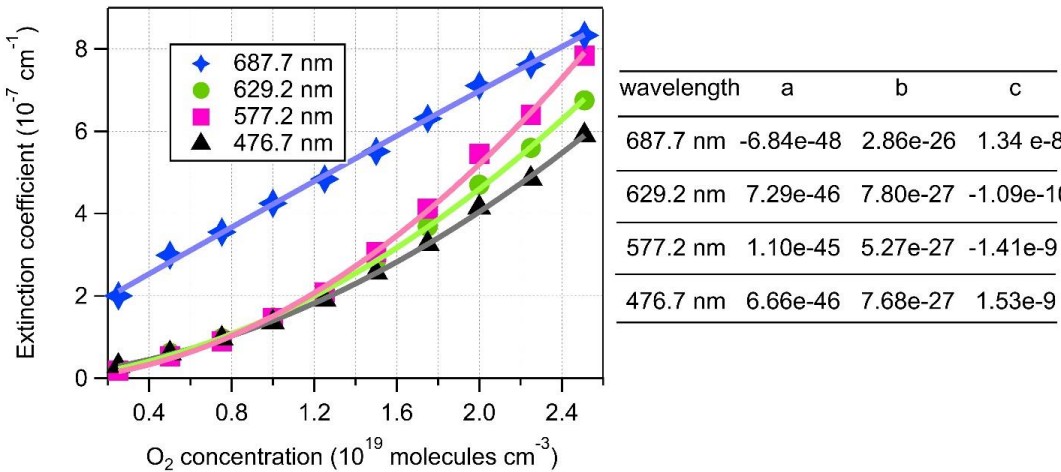


Figure 4. 2$^{rd}$ order polynomial fit of extinction coefficients measured by the BBCES. The $O_2$
concentration-dependent extinction coefficients are contributed by the extinction coefficients of
$O_2$ ($\sigma_{O_2}$), He ($\sigma_{He}$), and the $O_2$-$O_2$ CIA cross-sections ($\sigma_{O_2-O_2}$).





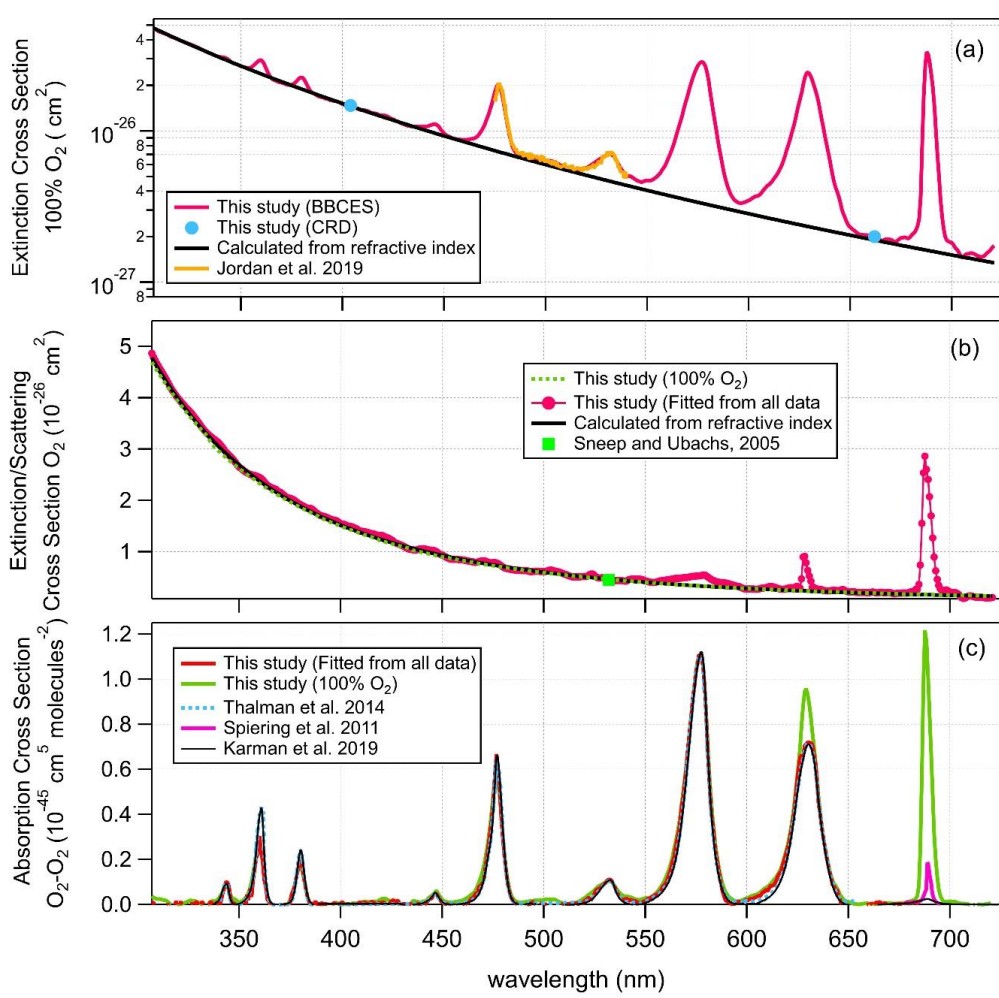


Figure 5. Wavelength-dependent extinction cross-sections of 100% $O_2$ (a), extinction cross-sections of $O_2$ (b), and $O_2$-$O_2$ CIA cross-section (c).




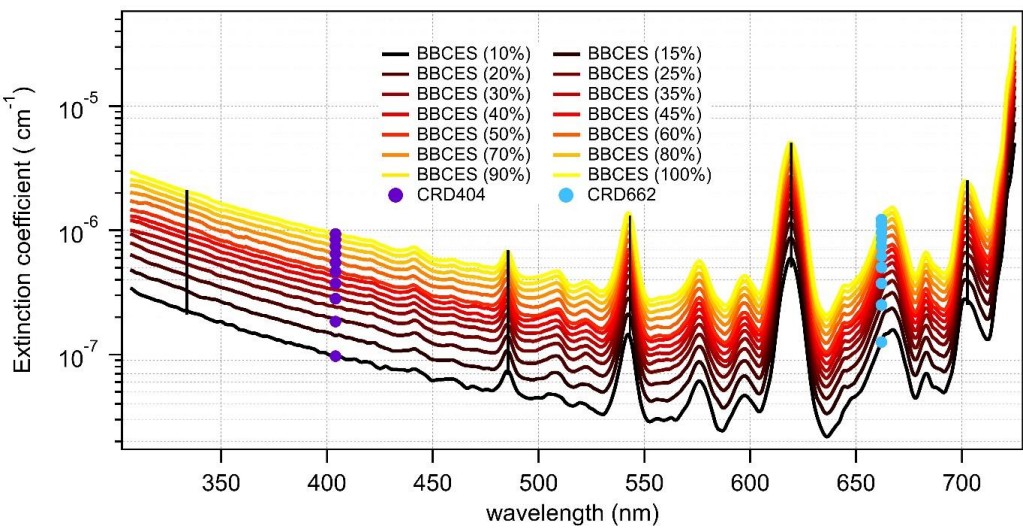

Figure 6. Wavelength-dependent extinction coefficients of $CH_4$ + He mixtures as a function of $CH_4$ mixing ratio. The colored lines represent extinction coefficients obtained from BBCES and markers represent results from CRDS. Measurements were performed with $CH_4$ percentage within 10% and 100% with a 10% step. Moreover, BBCES measurements were also performed for 15%, 25%, 35%, and 45% $CH_4$. The number concentration of 100% methane was $2.50143 \times 10^{19}$ molecules $cm^{-3}$. Data at selected wavelengths (vertical lines) are shown in Figure 7.





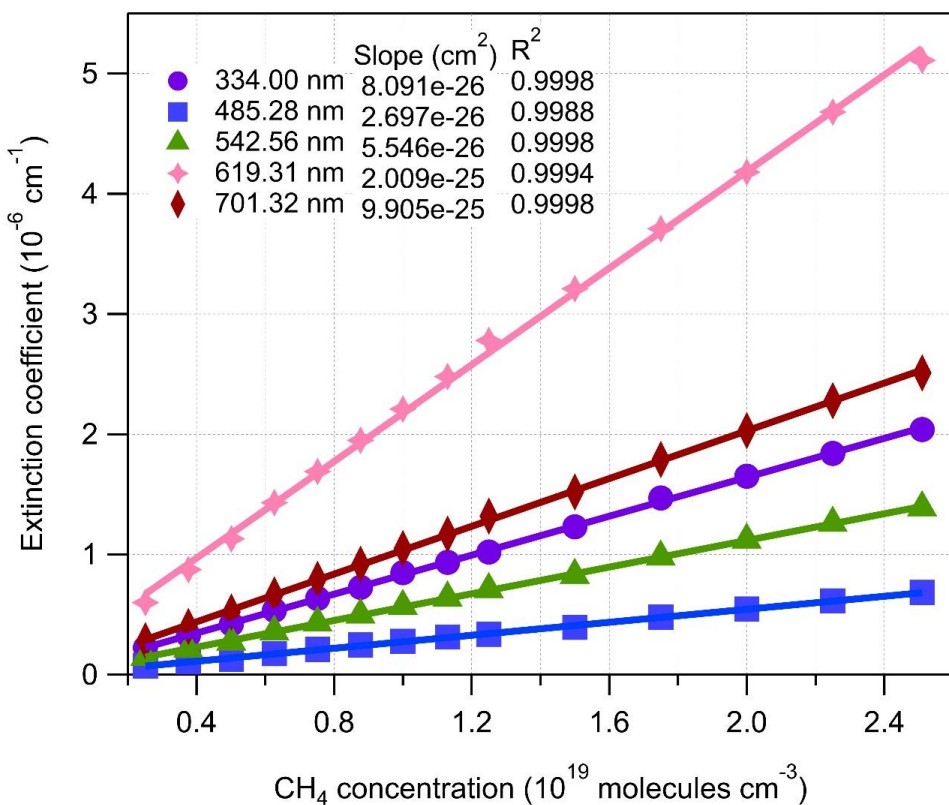


Figure 7. The relationship between BBCES measured extinction coefficients of $CH_4$+He mixtures and $CH_4$ concentration. The selected wavelengths were located in Figure 6 by vertical lines.



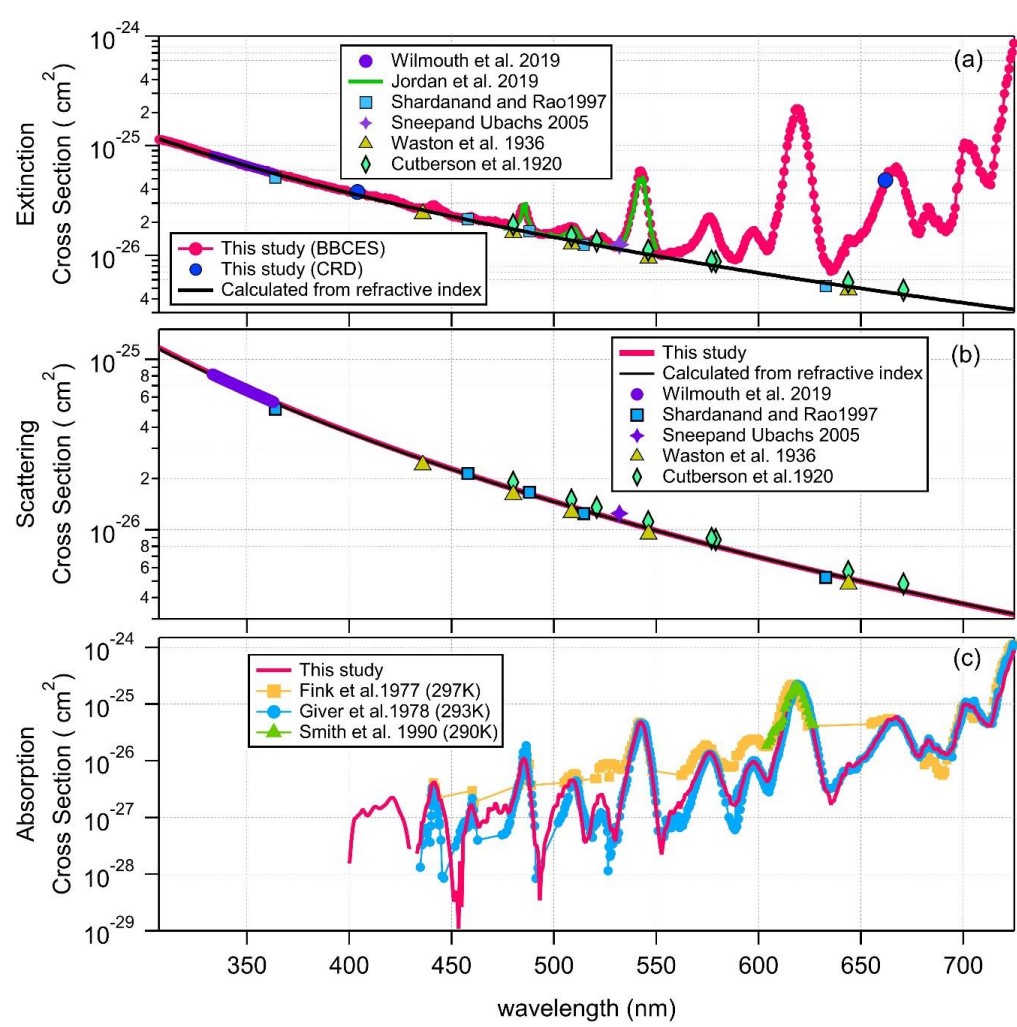


Figure 8. Extinction (a), scattering (b), and absorption (c) cross-sections of $CH_4$.

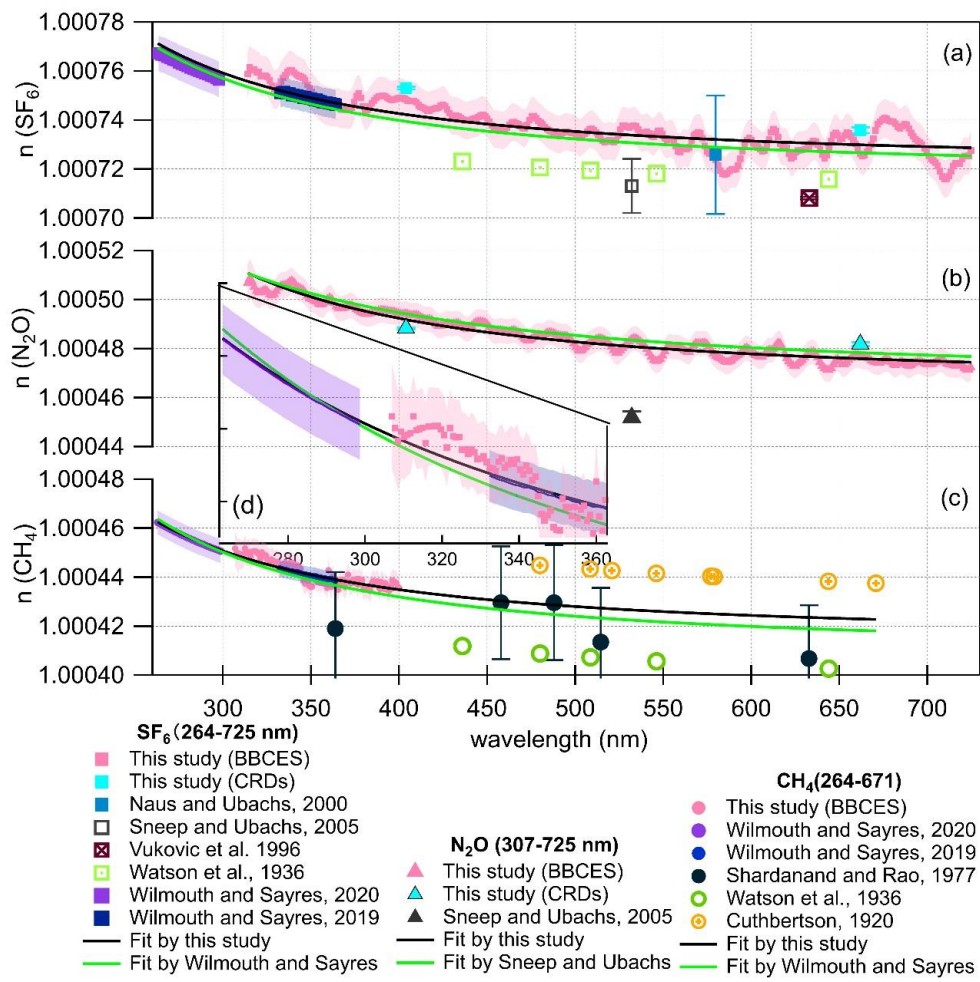

Figure
9. Real refractive index (*n*) for SF₆ (a), N₂O (b), and CH₄ (c). Comparison of Refractive index
from this work with previous studies (Cuthbertson and Cuthbertson, 1920; Naus and Ubachs, 2000;
Shardanand and Rao, 1977; Sneep and Ubachs, 2005; Vukovic et al., 1996; Watson et al., 1936;
Wilmouth and Sayres, 2019, 2020) over the wavelength range of 264–725 nm. The green line
represents the dispersion relation given in Table 1. The black line represents the dispersion relation
given in Eq. (8–10) derived from a fit to our data and references results. The *n* values for
Shardanand and Rao (1977), Sneep and Ubachs (2005), Naus and Ubachs (2000) were calculated
from their reported Rayleigh scattering cross-sections. Refractive index data from Sneep and
Ubachs (2005) are not used in the fitting since these results are away from others. Panel (d) is a
close-up view of the panel (c) in the wavelength range of 264–363 nm.



Table 1. Refractive index and King correction factors for calculating Rayleigh scattering cross-
sections and available measurements in the wavelength range of 300–725 nm. Measurements for
He and $N_2$ are not summarized in this table.

| Gas | Refractive index and King correction factors | | | | Measurements | |
|---|---|---|---|---|---|---|
| | $(n-1) \times 10^8$ | $F_k(\nu)$ | $\nu$ (cm$^{-1}$) | References | $\lambda$ (nm) | References |
| He | $2283 + \dfrac{18102 \times 10^{13}}{1.5342 \times 10^{10} - \nu^2}$ | 1.0 | 14285-33333 | Thalman, 2014; Leonard, 1974; Cuthbertson, 1932 | | |
| $N_2$ | $5677.465 + \dfrac{318.81874 \times 10^{12}}{1.44 \times 10^{10} - \nu^2}$ | $1.034 + 3.17 \times 10^{-12} \nu^2$ | 21360-39370 | Sneep, 2005; Naus, 2000 | | |
| $CO_2$ | $1.1427 \times 10^{11} \times \left( \dfrac{5799.25}{(128908.9)^2 - \nu^2} + \dfrac{120.05}{(89223.8)^2 - \nu^2} + \dfrac{5.3334}{(75037.5)^2 - \nu^2} + \dfrac{4.3244}{(67837.7)^2 - \nu^2} + \dfrac{1.218145 \times 10^{-5}}{(2418.136)^2 - \nu^2} \right)$ | $1.1364 + 2.53 \times 10^{-11} \nu^2$ | 39417-55340 | Alms, 1975; Bideau-Mehu, 1973; Sneep, 2005 | 333-725 | Jordan, 2019; Shardanand, 1977; Sneep, 2005; Wilmouth, 2019; He, 2018 |
| $CH_4$ | $4869.8 + \dfrac{4.1023 \times 10^{14}}{1.133 \times 10^{10} - \nu^2}$ | 1.0 | 15385-40000 | Sneep, 2005; Wilmouth, 2020 | 333-363, 434-725 | Cuthbertson 1920; Jordan, 2019; Shardanand, 1977; Sneep, 2005; Watson, 1936; Wilmouth, 2019; Smith, 1990; Giver, 1978; Fink, 1977 |
| $N_2O$ | $46890 + 4.12 \times 10^{-6} \nu^2$ | $\dfrac{3.3462 + 70.8 \times 10^{-12} \nu^2}{2.7692 - 47.2 \times 10^{-12} \nu^2}$ | 15453-21838 | Alms, 1975; Sneep, 2005 | 300-320, 532 | Johnston, 1975; Sneep, 2005 |
| $SF_6$ | $18611.4 + \dfrac{8.9566 \times 10^{14}}{1.680 \times 10^{10} - \nu^2}$ | 1.0 | 15385-40000 | Sneep, 2005; Vukovic, 1996; Wilmouth, 2020 | 333-363, 532, 633 | Sneep, 2005; Vukovic, 1996; Wilmouth, 2019 |
| $O_2$ [a] | $20564.8 + \dfrac{2.480899 \times 10^{13}}{4.09 \times 10^9 - \nu^2}$ | $1.09 + 1.385 \times 10^{-11} \nu^2 + 1.448 \times 10^{-20} \nu^4$ | 18315-34722 | Hohm, 1993; Sneep, 2005 | 328-667 | Thalman, 2013; Jordan, 2019; Hermans, 1999; Greenblatt, 1990; Spiering, 2011 |

Unless noted, the refractive index is scaled to 288.15 K and 1013.25 hPa. N = 2.546899 $\times 10^{19}$
molecules cm$^{-3}$.
Due to limited space, only the first name of each reference is shown in the table.
[a] The refractive index was obtained at 273.15 K and 1013.25 hPa, N = 2.68678 $\times 10^{19}$ molecules
cm$^{-3}$ is used in Eq. (1)





Table 2. The Rayleigh scattering cross-sections ($10^{-27}$ cm$^2$) calculated from the refractive index ($n$-based) and obtained from BBCES (Exp) of selected wavelengths.

| $\lambda$(nm) | CO$_2$ | | SF$_6$ | | N$_2$O | | O$_2$ | | CH$_4$ | |
|---|---|---|---|---|---|---|---|---|---|---|
| | $n$-based | Exp | $n$-based | Exp | $n$-based | Exp | $n$-based | Exp | $n$-based | Exp |
| 330 | 98.22 | 96.8 | 241.5 | 239.4 | 137.9 | 136.7 | 34.71 | 35.1 | 84.12 | 85.3 |
| 404 | 41.67 | 41.6 | 104.5 | 105.7 | 57.71 | 57.9 | 14.57 | 14.8 | 35.57 | 35.9 |
| 532 | 13.32 | 13.3 | 33.92 | 34.1 | 18.19 | 18.3 | 4.642 | 4.55 | 11.34 | 11.3 |
| 660 | 5.516 | 5.52 | 14.16 | 14.2 | 7.483 | 7.47 | 1.924 | 1.95 | 4.693 | 4.68 |
| 710 | 4.101 | 4.08 | 10.55 | 10.4 | 5.551 | 5.48 | 1.430 | 1.41 | 3.487 | 3.47 |



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
