# Peer review of "Scattering and Absorption Cross Sections of Atmospheric Gases in the 1"

_Atmospheric Chemistry and Physics, 2020_

## Referee Comment (RC1) · Anonymous Referee #1 · 4 Dec 2020

The authors report results from a laboratory study using Broadband Cavity Enhanced Spectroscopy (BBCES), supplemented by Cavity Ring Down Spectroscopy (CRDS) to measure Rayleigh scattering cross sections and absorption cross sections, where applicable, for the gases $CO_2$, $N_2O$, $SF_6$, $O_2$, and $CH_4$. What's new here is the use of a single mirror to cover a very broad wavelength range for the BBCES studies from 338-725 nm. The topics covered are important and within the scope of ACP, but I have significant concerns that need to be addressed before this paper can potentially be published.

— Major Comments/Corrections:

[Figure]

(1) The paper makes claims of superiority over prior work that do not appear to be supported by the data. For example, this sentence in the Abstract that is echoed in various other forms in the paper: "This study provides improved refractive index dispersion relations, n-based Rayleigh scattering cross-sections, and absorption cross-sections for these gases." In a few instances, the data from this paper do represent clear improvements over what is already in the literature (e.g., the significantly expanded wavelength range for the N2O Rayleigh scattering cross sections), but in many cases, the data from this paper can only be stated to be in agreement with past work within the stated uncertainties rather than "improved". The authors should carefully go through the manuscript and make sure they are not overstating the superiority of the present results over past studies.

This concern is especially relevant for the revised dispersion relations for SF6 and CH4. Based on the large spread in the n values from this study (and the range of pink shading, which I assume is the 1-sigma uncertainty) in Figure 9a and 9c, it seems clear that the data from this study do not provide constraint on the dispersion relations to allow distinguishing between the black and green fit lines. Moreover, the Shardanand and Rao data, which have large uncertainties and were intentionally excluded from analyses in earlier studies, only serve to increase the uncertainty of the CH4 fit here rather than decrease it. There are other issues with Figure 9 to be addressed later, but the point here is that the inclusion of the pink points from this study and black dots from Shardanand and Rao do not provide improved dispersion relations for SF6 and CH4 over the existing n-based relations. I find it disconcerting that nearly all of the pink points at wavelengths shorter than 480 nm lie above the fit line for SF6. Similarly, for CH4 nearly all of the pink points shorter than 400 nm are above the fit line except in the region 340–360 nm, where all points lie below the fit line. There appear to be systematic biases as a function of wavelength. Moreover, there is no discussion in Section 3.5 of how the different data sets were weighted to produce the fits.

In short, I do not see anything convincing in the manuscript that the dispersion relationships produced for SF6 and CH4 are "improved" relative to the latest n-based values in the literature, and it is very probable that the relations here are less accurate (see below). I recommend taking a consistent approach for SF6 and CH4 as to that used with CO2, which is to simply say that your results agree with the existing n-based values in the literature, rather than unnecessarily providing new dispersion relationships for SF6 and CH4 that cannot be distinguished from current values within your uncertainty.

(2) The accuracy of the Rayleigh scattering cross sections appears to be overstated. In section 2.4, the pressure is stated to be accurate to 0.01% and temperature to 0.1%. While the authors likely have the precision to measure changes in P and T to these levels, this level of accuracy would be surprising. Most pressure gauges are not even capable of 0.01% accuracy. How and where were P and T measured? How were the P and T gauges calibrated to this accuracy? Please provide part numbers for all devices used in the experiments.

More importantly, the authors appear to be claiming that they have determined the reflectivity of the mirrors with essentially no experimental uncertainty. The 1.03% stated uncertainty in section 2.4 – which is the uncertainty in the effective pathlength, not the mirror reflectivity (please correct this misstatement in line 231) – is almost entirely due to the 1.0% N2 reference cross section. But what about the standard deviation of the N2/He runs? Surely, there was some spread in the data from run to run? And there undoubtedly was a wavelength dependence in the data quality based on mirror reflectivity, mirror transmission, fluctuations in the light source, the amount of light to the detector, etc.

There is an additional problem here in that the reference N2 cross section in Table 1 is only valid up to 468 nm. At longer wavelengths, the 1% error should be increased to account for the uncertain extrapolation of this dispersion relation.

Most concerning of all with respect to stated uncertainties is that for the individual gases, the authors appear to be defining the 1-sigma uncertainty in their results based

on the differences between their values and the existing n-based values. If this was indeed the approach taken, this is wrong and must be corrected. If this was not the approach taken, the method used should be clarified. Lines 262–264 simply state uncertainty values of 1.5%, 1.1%, and 1.5% for CO2, N2O, SF6, respectively, without explanation where they came from – these numbers appear again in the Conclusions Line 434 when these exact same numbers are stated to be the relative difference with the n-based calculations. As I examine the data plotted in Figure 1d, it is clear that the precision of the data in this study is generally quite good. But Figure 1e reveals that the accuracy is substantially worse than the precision. There is nearly +4% to –4% discrepancy for CO2 and SF6 at some wavelengths relative to the n-based values. Whether or not the n-based values are correct is not the point here – the point is that the n-based values are smooth – so all of this chatter is due to the experimental data in the present paper. For CO2 and SF6, there appears to be a 7% swing just in going from around 590 to 600 nm. And this large error is in the mean values – the variation would be even greater in the individual runs. I suspect that the highly structured reflectivity curve (Figure S1) is creating problems, i.e., systematic biases creating persistent inaccuracies across the wavelength range even though the data reproducibility (precision) is good.

In short, for all the reasons mentioned above, the uncertainties should be completely revisited, more thoroughly explained, and almost certainly increased significantly from the present values.

(3) The authors should cite previous studies where appropriate. Reference to the n-based values is made repeatedly throughout the paper for various gases, but it is never stated clearly from which study each of these n-based values originated. I recommend in column 5 of Table 1 that it be highlighted in some way (e.g., asterisk, bold, footnote) which of the listed References the n-based expression in column 2 came from. Also, the source of the n-based expression should be named for each gas at least once in the text and when comparisons are being made.

It would seem appropriate after Lines 80-81 to mention that Wilmouth and Sayres 2020 just published SF6 and CH4 dispersion relationships for the wavelength range 250-650 nm. Lines 90-93, 392-398, and 412-417 all discuss a method of determining new dispersion relationships for SF6 and CH4 in this study that is a copy of the Wilmouth and Sayres 2020 method, but the Wilmouth and Sayres 2020 paper is not credited for the idea that this paper is replicating.

(4) Figure 9 contains significant errors. The n-based data in Figure 9 for the studies of Watson, Cuthbertson, and Vukovic are all plotted incorrectly, causing them to appear much further from the fit lines than they should be. The authors appear to have taken the index of refraction data directly from the papers and did not appropriately convert to the reference temperature of 288.15 K.

The authors should also check Figure 8 for accuracy, where the same reference studies are plotted.

(5) Due to the errors highlighted in point (4) above, the dispersion relationships produced in this study that included the literature index of refraction data are therefore all incorrect. The authors also do not state how they converted their own Rayleigh scattering data to index of refraction, nor the temperature and pressure at which their dispersion equations were defined; these things should be addressed.

Again, once the data themselves are plotted correctly, I see no point in producing new dispersion formulas for SF6 and CH4 that cannot be distinguished from the existing dispersion formulas within uncertainty. Simply state that the data here agree with existing values, as was done with CO2.

— Additional Comments/Corrections:

Line 1 and elsewhere: Cross-sections should not be hyphenated

Lines 60–62: It is typical to order references in a list like this chronologically. Add Thalman et al. 2017 (Erratum to 2014 paper) and Wilmouth and Sayres 2020 to this

list.

Line 98 and many other places in the manuscript: The authors use "BBCES" as if it were the name of their experimental setup. If the sentence doesn't make sense with "Broadband Cavity Enhanced Spectroscopy" substituted in place of "BBCES", the sentence should be edited.

Lines 104-120: Please provide more detail on how the broad wavelength range was filtered and how the data acquisition worked. Did the spectrometer scan through the wavelength range X nm at a time? Were 3 seconds of data acquired at each wavelength per spectrum, or it took 3 seconds to scan the entire range? With light being present over such a broad wavelength range, how was stray light prevented from impacting the results? It seems like this would have been a problem. For example, did you ever use a longpass filter to block the shorter wavelength light to see if there was still signal present at the detector at the shorter wavelengths due to stray light from the longer wavelengths?

Line 129: Why not present this equation already solved for R?

Section 2: There is an awkward switching between past and present tense throughout. Pick one (ideally past) and stick with it.

Line 155 and many other places: The authors use "CRDS" as if it were the name of their experimental setup. If the sentence doesn't make sense with "Cavity Ring Down Spectroscopy" substituted in place of "CRDS", the sentence should be edited.

Line 272: Delete "theoretical". The n-based calculations are based on experimentally determined data.

Line 345: Shouldn't 630 nm here be 629 nm, per Line 321?

Lines 346-348: Some of these discrepancies appear quite large, and the explanation given that it is because they are the lowest intensity bands is not entirely true. The band at 532 nm appears smaller than the band and 360 and 380 nm, and yet it agrees well.

What the bands with larger discrepancies have in common is not their size, it's that they are all located at wavelengths below 450 nm, while the ones that agree well are all located at wavelengths above 450 nm. Please comment on why this wavelength-dependent disagreement might be the case. The text says Thalman and Volkamer 2013 are being plotted, but the legend in the figure says Thalman et al. 2014. Also, why is there no comparison in the text with the most recent Karman 2019 study?

Line 355-356: Please add a reference for this first sentence.

Lines 371-373: This sentence is incorrect as written. The interferometer studies obtained index of refraction values, not scattering cross sections.

Line 383: It seems like an oversimplification to say that there is "good" agreement here. There are significant discrepancies at many wavelengths.

Line 391: Delete "much".

Line 392: "Better constrain" relative to what?

Line 412: Change 9b to 9c

Lines 429-432, Lines 361-363, and Lines 327-332: The BBCES and CRDS agreement is certainly a positive result, but be careful not to overstate what this means (lines 329-331). It's just two wavelengths, and there are clearly significant wavelength-dependent errors in the BBCES data. Moreover, the locations of the CRDS points are all offset from both the fit lines and the BBCES data in Figure 9, suggesting that the CRDS results do not provide any additional constraint on the accuracy of the BBCES results even at these two wavelengths.

Line 442-445: This point was made earlier, but to reiterate here, the claims of superiority from this study are not justified. The Wilmouth and Sayres 2020 fit of the UV region for CH4 that is plotted in Figures 9c and 9d was purposefully weighted such that it better represented the existing index of refraction data in the visible region at the slight expense of the UV fit, as described in that paper. A UV-only fit is also presented

in that paper. The fact that the fit from the present manuscript is higher for CH4 simply means that this new fit is not representing the visible data as well as the Wilmouth and Sayres 2020 expression.

Figure 8: There are many errors in the figure legends. Wilmouth, Watson, and Cuthbertson are all two-author papers, not "et al." Watson is misspelled. Cuthbertson is misspelled. Add a space after Rao. Add a space after Sneep.

Figure 9: It is never stated what the shading represents in this figure.

Table 1: The He equation is incorrect. There is a missing decimal in 1.8102.

— Minor Corrections:

Line 22: Delete "and" after 1.5%

Line 31: Delete "as" after either

Line 44: Add "a" after For

Line 53: Change "index" to indices

Line 151: Change "systems" to spectroscopy

Line 152: Change "-validation of" to validate

Line 156: Add "was" after thus

Line 209: Change "overlaps" to overlap

Line 216: Delete "the" after of

Line 235: Change "resulted uncertainty of" to resulting uncertainty in

Lines 306 and 313: Avoid apostrophe "instrument's" – instead say wavelength resolution of the instrument

Line 328: Change "slops" to slopes

---

## Referee Comment (RC2) · Andreas Richter (Referee) · 13 Jan 2021

**Initial review of "Scattering and Absorption Cross-sections of Atmospheric Gasses in the Ultraviolet-Visible Wavelength Range (307 – 725 nm)" by Quanfy He, Zhen Fang, Ofir Shoshamin, Steven S. Brown and Yinon Rudich.**

The research presented in the manuscript is interesting, and certainly relevant to the journal Atmospheric Chemistry and Physics. it is somewhat unfortunate that nitrogen is used for the calibration of the mirror reflectance, as nitrogen itself is a highly interesting gas for these type of measurements. However, the manuscript needs to be improved significantly before it is ready for publication.

I have doubts about the direct measurements of the oxygen absorption bands presented rather prominently in the abstract (the delta and gamma overtone bands and the B-band) should be included at all. The spectral resolution of the instrument is simply not suitable to produce a meaningful result for these bands. I interpreted the "0.8 nm resolution" statement as a FHWM value, and produced a plot of high resolution cross sections of oxygen (HITRAN; modelled concentration is $21\,\%$ $O_2$ in $N_2$ at $1018\,\text{hPa}$ and $294\,\text{K}$ and includes pressure and temperature broadening) and overplotted a Gaussian spectral response function with FWHM of 0.8 nm, see figure 1. This combination cannot produce a meaningful result. The range of absorption cross sections spans several orders of magnitude within the spectral response of the instrument. In a CRDS setup this would lead to a multi-exponential decay, an underdetermined problem. I'm not sure the situation of interpreting the output of a BBCES instrument when a wide range of absorptions is present within a spectral pixel is any better. At best the result depends on the spectral resolution, making it far less useful to others. Also note that this is before taking any broadband collision induced effects into account, but that is probably the least of your worries in this case. Taking out these three figures leaves enough material for an article that is worth publishing, so I do not see this as a significant drawback or a hindrance to the publication of the remaining results. The other features ($O_2$-$O_2$ absorption and Ralyeight scattering) have spectral features that are much wider than 0.8 nm, and as such can be investigated with this instrument. I'm not enough of an expert to judge the methane absorption features in this spectral range, but I would consider these suspect as well.

This means a substantial rewrite of section 3.3, to reduce the section on how the $O_2$-$O_2$ absorption was isolated from the oxygen absorption and Rayleigh losses. Please note that at line 286 it is mentioned that the extinction cross section of the oxygen monomer is linearly correlated with the oxygen concen-

[Figure]

Figure 1: Oxygen B-band absorption cross sections from HITRAN. The modelled concentration is $21\,\%$ $O_2$ in $N_2$ at $1018\,hPa$ and $294\,K$ and includes pressure and temperature broadening. In red a Gaussian spectral response function with FWHM of 0.8 nm in shown.

tration. Be very careful with that statement, as the wings of the oxygen lines also show a $p^2$ dependence due to pressure broadening.

A the end of section 3.3 there is a contradiction: on line 326 the 630 nm is listed as within 1.1 % of Thalman and Volkamer (2013), while on line 333 the same wavelength is listed as "this method cannot derive the cross-sections of CIA of $O_2$-$O_2$ at 630 and 688nm". Either alter the statement or explain better what is going on here.

I should note there that I don't have access to He et al., 2018, so some of my remarks may have been covered there. However, some details on the experimental setup should be mentioned briefly here as an aid to the reader. I'll indicate what I'd like to see added in the technical remarks below.

In the method section a lot of the equipment is mentioned. I'm missing some crutial information on the pressure sensor, the temperature probe and how the gas was mixed at specific mixing ratios, and the error introduced there. I do not see any discussion of the wavelength calibration. For the CRDS this is likely inherent in the used source (not mentioned here either) but for the BBCES the calibration method for the spectrometer should at least be mentioned.

In section 2.2 I'm missing some details on the CRDS technique. There are a lot of methods to initiate a ring down of an optical cavity, and this details is not in the description. The reference (Bluvshtein et al., 2012) uses a 20 Hz Nd:YAG laser, whereas here the manuscript reads: "Over 1000 decay time measurements are monitored and averaged on a second basis", suggesting that a much faster laser system was employed here.

In the conclusion (line 446 to 449) a reference is made to a wavelength range that is not part of this study at all. On line 441 nitrogen is explicitly mentioned as a gas of interest for atmospheric observations in particular for Lidar analysis, also including a wavelength that is outside the scope of the article. While I appreciate the outlook for future studies, please explicitly mark these statements as such, as they are not part of the conclusions of this study.

Finally, the choice of colours hinders accessibility for many colourblind people. There are combinations of colours for use in figures that will make the results more readable for colourblind people. Printing in black and white will quickly show where the use of colours should be improved.

**1   Specific remarks**

Line 43: please refer to the original year of publication in the reference (1899), not the year of the re-issue of the collected papers. Anyone familiar with the subject matter will be confused by Strutt 2009, but at the same time know immediately what Strutt 1899 is.

Line 50 (and several other locations throughout the manuscript): really weird line breaks.

Line 74: Nitrogen should have subscript "2" not "e".

Line 99 in the Methods section: I found Bluvshtein et al., 2016 (doi:10.5194/amt-9-3477-2016) to be the end point of a chain of references for the method that starts with the two that are mentioned. I suggest to use this reference (in addition or instead of). Sending readers into a rabbit hole to chase the methods is not nice.

Line 226: In the results and discussion I read: "The reflectivity of the cavity mirrors, measured across the entire range using the difference in Rayleigh scattering of $N_2$ and He, was very stable throughout the experiments". I expect to find this in the method section, how this was determined.

Line 231: the reflectivities and the losses have their relative order interchanged, please maintain oder for readability.

Line 285: It is worthwhile to note what transitions of the $O_2$-$O_2$ absorption features occur here, and especially that both molecules leave the interaction in an electronically excited state, at least for the shorter wavelengths.

**2    Figures and captions**

Line 466: Caption does not match labels in figure. Given the magnitude of the signal in the figure, the error is in the caption, not the figure.

Figure 3: missing units on the axes.

Figure 4: consider listing tabular material in a table rather than a figure legend.

Figure 5: the unit of panel (c) cannot be correct, there must be a density involved here.

Figure 6: the methane percentage is mentioned, but not the cell density for $100\,\%$ methane concentration.

Figure 6 & 7: suggest to add markersto figure 6 indicating where the wavelengths are that are shown in figure 7.

Figure S3: missing units on the axes

Figure S4: in black  white the traces look identical.

---

## Editor Comment (EC1) · Andreas Richter (Editor) · 13 Jan 2021

Attached you will find the author's reply to the initial review. Please note that the comments and replies are for an earlier version of the manuscript, and that the preprint which is available on this site has already incorporated the revisions discussed here by the authors.

Please also note the supplement to this comment:
https://acp.copernicus.org/preprints/acp-2020-941/acp-2020-941-EC1-supplement.pdf

[Figure]

[Figure]

**Supplement:**

**Referee 2**

Initial review of "Scattering and Absorption Cross-sections of Atmospheric Gasses in the Ultraviolet-Visible Wavelength Range (307 -725nm)" by Quanfu He, Zhen Fang, Ofir Shoshamin, Steven S. Brown and Yinon Rudich.

The research presented in the manuscript is interesting, and certainly relevant to the journal Atmospheric Chemistry and Physics. it is somewhat unfortunate that nitrogen is used for the calibration of the mirror reflectance, as nitrogen itself is a highly interesting gas for these type of measurements. However, the manuscript needs to be improved significantly before it is ready for publication.

Reply: We thank the reviewer for these important comments. These comments helped a lot to improve our manuscript. We agree with the reviewer that nitrogen by itself a highly interesting gas for Rayleigh scattering cross-section measurement. However, we need one gas that has significantly larger extinction cross-sections than He to calibrate our BBCES. The previous study by Thalman et al. (2013) has pointed out that their CRD measurement agrees with the *n*-based calculation within 1%. It is noted that  $N_2$  is widely used for the calibration of BBCES to measure extinction/scattering/absorption of various gases.

I have doubts about the direct measurements of the oxygen absorption bands presented rather prominently in the abstract (the delta and gamma overtone bands and the B-band) should be included at all. The spectral resolution of the instrument is simply not suitable to produce a meaningful result for these bands. I interpreted the 0.8 nm resolution statement as a FHWM value, and produced a plot of high resolution cross sections of oxygen (HITRAN; modelled concentration is 21% O2 in N2 at 1018hPa and 294K and includes pressure and temperature broadening) and overplotted a Gaussian spectral response function with FWHM of 0.8 nm, see figure 1. This combination cannot produce a meaningful result. The range of absorption cross sections spans several orders of magnitude within the spectral response of the instrument. In a CRDS setup this would lead to a multi-exponential decay, an underdetermined problem. I'm not sure the situation of interpreting the output of a BBCES instrument when a wide range of absorptions is present within a spectral pixel is any better. At best the result depends on the spectral resolution, making it far less useful to others. Also note that this is before taking any broadband collision induced effects into account, but that is probably the least of your worries in this case. Taking out these three figures leaves enough material for an article that is worth publishing, so I do not see this as a significant drawback or a hindrance to the publication of the remaining results. The other features (O2-O2 absorption and Rayleigh scattering) have spectral features that are much wider than 0.8 nm, and as such can be investigated with this instrument. I'm not enough of an expert to judge the methane absorption features in this spectral range, but I would consider these suspect as well.

This means a substantial rewrite of section 3.3, to reduce the section on how the  $O_2$ - $O_2$  absorption was isolated from the oxygen absorption and Rayleigh losses.

Figure 1: Oxygen B-band absorption cross sections from HITRAN. The modelled concentration is 21%  $O_2$  in  $N_2$  at 1018hPa and 294K and includes pressure and temperature broadening. In red a Gaussian spectral response function with FWHM of 0.8 nm in shown.

Reply: We thank the reviewer for this constructive comment. We agree with the reviewer that due to the low instrument wavelength resolution and the discrete structured of the B,  $\gamma$ , and  $\delta$  bands, the absorption cross-sections measured for these bands are less useful to others. Thus we accepted the Reviewer's suggestion and deleted all of the results regarding O2 absorption bands. We also revised the abstract and the conclusion section. It is noted, though that the data for the broad unstructured O2-O2 collision-induced absorption bands and the Rayleigh scattering cross-sections are still valid and useful. The agreement between our results (Rayleigh scattering and O2-O2 absorption cross-sections) and literature data validates our method. We prefer to keep this information in the manuscript. Detailed changes are as follows,

We added the following text in Section 3.3 "These absorption bands can only be resolved by a high-resolution spectrascopic technique. Absorption cross-sections of the B,  $\gamma$ , and  $\delta$  bands were convoluted from the HITRAN database (Gordon et al., 2017) by considering the temperature, pressure, and instrument's wavelength resolution. The wings of the oxygen lines also show a quadratic dependence on the pressure due to pressure broadening. However, due to the minimal O2 absorption contribution below 680 nm and the low instrument wavelength resolution, the extinction cross-section of the O2 monomer can be treated as linearly correlated with the O2 concentration. Moreover, the extinction cross-section of the O2 monomer is linearly correlated with the Square of the O2 concentrations. Moreover, these cross-sections can be retrieved from measurements at different O2 concentrations. Moreover, the V2–O2 CIA cross-section is correlated with the square of the O2 concentrations. Due to the discrete structured absorption bands and the instrument's wavelength resolution, the range of absorption cross-section spans several orders of magnitude within the spectral response of the instrument, limiting the relevance of the absorption crosssections for other researchers. These results are not further discussed here. However, the data for broader, unstructured CIA of  $O_2$ - $O_2$  are still useful for various applications. (Line 308-322)

We deleted the following sections, "Absorption cross-sections of O2 absorption bands were measured directly at 579 nm  $(1.8\pm0.3\times10^{-27} \text{ cm}^2)$ , 629 nm  $(6.4\pm0.3\times10^{-27} \text{ cm}^2)$ , and 688 nm  $(2.69\pm0.08\times10^{-26} \text{ cm}^2)$ ." (Line 23-24), "but positive values for O2 B band absorption" (Line 345), "The extinction cross-section data between 307–550 nm and 650–680 was further used to derive the refractive index of O2 and then extrapolate it to the entire wavelength range for calculating the Rayleigh scattering cross section. The absorption cross-section was then calculated as the difference between the extinction cross-section and Rayleigh scattering cross-section. Three absorption peaks corresponding to the molecular oxygen B band at 688 nm,  $\gamma$  overtone band at 629 nm, and  $\delta$  overtone band at 580 nm were found and the determined absorption cross-sections at their center wavelengths are  $(2.69\pm0.08)\times10^{-26}$ ,  $(6.4\pm0.3)\times10^{-27}$ , and  $(1.8\pm0.3)\times10^{-27}$  cm2, respectively. The absorption cross-sections contribute 81%, 24%, and 8.0% to the extinction of each wavelength. These fractions are consistent with the estimation by Thalman and Volkamer (2013)." (Line 351-360), and "Absorption cross-sections of the molecular oxygen bands ( $\delta$ ,  $\gamma$  and B) were derived directly at 579 nm  $(1.8\pm0.3\times10^{-27} \text{ cm}^2)$ , 629 nm  $(6.4\pm0.3\times10^{-27} \text{ cm}^2)$ , and 688 nm  $(2.69\pm0.08\times10^{-26} \text{ cm}^2)$ , respectively." (Line 452-454)

There is no available high-resolution spectroscopy data for CH4 below 869 nm from the HITRAN database. Previous studies by Fink et al. (1977), Giver et al. (1978), and Smith et al. (1990), have determined the absorption bands of CH4 at wavelength resolution of 1, 0.05-0.06, 0.025 nm, respectively. The latter two studies' wavelength resolution is sufficient for quantitative spectral studies of individual vibrational-rotational lines of CH4 (Giver et al. 1978, Smith et al. 1990). The observed absorption bands are smooth, unstructured, and the spectral features are substantially broader than 0.8 nm. Therefore, these absorption features can be investigated by our BBCES. Moreover, the BBCES measured extinction coefficients at 334, 485.28, 542.56, 619.31, and 701.32 nm that significantly linearly correlated with the CH4 concentration, with R2>0.9988. Our results also agree well with the literature results. Thus, we keep the results for CH4 in the manuscript. We add the following sentence "Previous high resolution spectroscopy studies have identified smooth and unstructured absorption bands across the UV-visible range (Giver, 1978; Smith et al., 1990). The spectral features are substantially broader than 0.8 nm, thus the absorption by CH4 can be measured by our BBCES."

Please note that at line 286 it is mentioned that the extinction cross section of the oxygen monomer is linearly correlated with the oxygen concentration. Be very careful with that statement, as as the wings of the oxygen lines also show a p2 dependence due to pressure broadening.

Reply: We thank the reviewer for this comment. The absorption from the  $\gamma$  overtone at 629 nm and the  $\delta$  overtone band at 580 nm contribute a fraction (<20%) to the total absorption. Our experiments use different mixing ratios of oxygen in He at a constant total pressure and temperature. The change in the wings of the O2 molecular absorption spectrum would therefore be the exchange of pressure broadening for O2 self broadening. The average pressure shifts measured for the B and  $\gamma$  bands are -0.0087 and-0.0095 cm-1 atm-1, respectively (Barnes and Hays, 2002). The self-broadening collision coefficients for the  $\gamma$  overtone have been determined

from the absorption line width and were found to vary from 0.055 to 0.037 cm-1 atm-1 (Mélières et al., 1985). The O2 partial pressure changed from 0.1 to 1 atm during our experiment while the totoal pressure is constant. The resulted broadening are withing the spectra resolution of this study. Thus the pressure broadening effect for wings of the oxygen lines has a small contribution to the strong O2-O2 CIA peak centered at these two wavelengths. As we can see from the results, the absorption cross-sections for Rayleigh scattering and O2-O2 CIA dericed by this method are in good agreement with literature data, thus our method provide reliable results. For data processing, the extinction cross-section can be treated as linearly correlated with the O2 concentration. We clarified this in the revised manuscript as "The wings of the oxygen lines also show a quadratic dependence on the pressure due to pressure broadening. However, due to the small contribution of O2 absorption below 680 nm and the low instrument wavelength resolution, the extinction cross-section can be treated as linearly correlated with the O2 concentration." (Line 311-315)

Barnes, J. E., and Hays, P. B.: Pressure Shifts and Pressure Broadening of the B and  $\gamma$  Bands of Oxygen, J. Mol. Spectrosc., 216, 98-104, https://doi.org/10.1006/jmsp.2002.8689, 2002.

Mélières, M. A., Chenevier, M., and Stoeckel, F.: Intensity measurements and self-broadening coefficients in the  $\gamma$  band of O2 at 628 nm using intracavity laser-absorption spectroscopy (ICLAS), J. Quant. Spectrosc. Radiat. Transf., 33, 337-345, https://doi.org/10.1016/0022-4073(85)90195-5, 1985.

At the end of section 3.3 there is a contradiction: on line 326 the 630 nm is listed as within 1.1% of Thalman and Volkamer (2013), while on line 333 the same wavelength is listed as this method cannot derive the cross-sections of CIA of  $O_2$ - $O_2$  at 630 and 688nm". Either alter the statement or explain better what is going on here.

Reply: As described in the methods section, two methods are used to derive the absorption crosssections of O2-O2 CIA. Method1: by performing  $2^{nd}$  polynomial fitt to the concentration-dependent extinction coefficients to get the absorption cross-sections of O2-O2 absorption (Line 197-205). Method2: Only using the extinction coefficient data from 100% O2 measurement (Line 206-214). We subtracted the scattering cross-section of O2 from the measured total extinction to get the absorption cross-sections. However, the O2 absorption bands at 580, 630, and 690 nm overlap with those of O2-O2 collisions. Thus only by using 100% O2 measurement, we can not derive the absorption cross-sections of O2-O2 at these wavelengths. In line 326, we describe the results from Method1, while in line 333, the results from Method 2 are described.

I should note there that I don't have access to He et al., 2018, so some of my remarks may have been covered there. However, some details on the experimental setup should be mentioned briefly here as an aid to the reader. I'll indicate what I'd like to see added in the technical remarks below.

In the method section a lot of the equipment is mentioned. I'm missing some crucial information on the pressure sensor, the temperature probe and how the gas was mixed at specific mixing ratios, and the error introduced there. I do not see any discussion of the wavelength calibration. For the CRDS this is likely inherent in the used source (not mentioned here either) but for the BBCES the calibration method for the spectrometer should at least be mentioned. Reply: The temperature sensor is a K-type thermocouple, and the pressure sensor is a Precision Pressure Transducer from Honeywell. The wavelength of the spectrometer was calibrated using an HG-1 Mercury lamp within the wavelength range of 296.728 and 738.393 nm. The way we mixed the gas was described in Line 147-149. The precision of the mass flow controllers is 0.5 mL min-1. When the total flow rate is 500 mL min-1, the uncertainty of the gas concentration (10-100%) varies from 0% to 1.0% (See table below). Specifically, when the gas concentration is within 20-80% or 100%, the uncertainty is below 0.5%. We included this uncertainty in the error propagation process for measurements of CH4+He and O2+He. We added this information in the revised manuscript.

Table 1. Uncertainty of the gas concentration introduced by mixing.

| Percentage (%)  | 10  | 15   | 20   | 25   | 30   | 35   | 40   | 45   | 50   | 60   | 70   | 80   | 90  | 100 |
|-----------------|-----|------|------|------|------|------|------|------|------|------|------|------|-----|-----|
| Uncertainty (%) | 1.0 | 0.68 | 0.52 | 0.42 | 0.36 | 0.32 | 0.30 | 0.29 | 0.28 | 0.17 | 0.36 | 0.13 | 1.0 | 0   |

Line 180-181 "The gas temperature (K-type thermocouple) and cavity pressure (Precision Pressure Transducer, Honeywell International Inc., MN, USA) were recorded for gas…"

Line117-119 "Before gas measurement, the wavelength of the spectrometer was calibrated using an HG-1 mercury argon calibration light source (Ocean Insight, USA) within the wavelength range of 302.15–727.29 nm."

Line238-241 "The precision of the mass flow controllers is 0.5 mL min-1. When the total flow rate is 500 mL min-1, the resulted uncertainty of the gas concentration (10-100%) varies from 0% to 1.0%. Thus, the overall 1- $\sigma$  uncertainty of extinction coefficients measured for CH4+He and O2+He varies from 1.1% to 1.5%."

In section 2.2 I'm missing some details on the CRDS technique. There are a lot of methods to initiate a ring down of an optical cavity, and this details is down of an optical cavity, and this details is not in the description. The reference (Bluvshtein et al., 2012) uses a 20 Hz Nd: YAG laser, whereas here the manuscript reads: Over 1000 decay time measurements are monitored and averaged on a second basis", suggesting that a much faster laser system was employed here.

Reply: The CRDS at 404 nm used in this study is similar to that described in Bluvshtein et al., 2012, and it is the same system shown in Bluvshtein et al., 2016. We are now using a 110mW diode laser (iPulse, Toptica Photonics, Munich, Germany) instead of a Nd:YAG laser as the light source. The diode laser is modulated to 1383 Hz at 50% duty cycle. We added more details of the CRDS in the revised manuscript.

"...measurement can be found in Bluvshtein et al. (2016) and He et al. (2018). Briefly, diode lasers (110 mW 404 nm diode laser, iPulse, Toptica Photonics, Munich, Germany; 120 mW 662 nm diode laser, HL6545MG, Thorlabs Inc., NJ, USA) are used as the light source of these CRDS. The 404 nm and 662 nm lasers are modulated at 1383 Hz and 500 Hz with a 50% duty cycle. The diode lasers are optically isolated by quarter waveplates ( $1/4 \lambda$ ) and polarizing beam splitters to prevent damage to the laser head by back reflections from the highly reflective CRDS mirror. The back-reflected light beam is directed into a photodiode, which serves as an external trigger source. Light

transmitted through the back mirror of the cavity is collected by an optical fiber and detected by a photomultiplier tube (PMT), which samples at a rate of 10 to 100 MHz. The time-dependent intensity data is acquired with a 100MHz card (PCI-5122, National Instruments, USA) and processed by a data acquisition software in Labview. An exponential curve is fitted to..." (Line 161-172)

In the conclusion (line 446 to 449) a reference is made to a wavelength range that is not part of this study at all. On line 441 nitrogen is explicitly mentioned as a gas of interest for atmospheric observations in particular for Lidar analysis, also including a wavelength that is outside the scope of the article. While I appreciate the outlook for future studies, please explicitly mark these statements as such, as they are not part of the conclusions of this study.

Reply: Thank you for this suggestion. We added one sentence at the end of the conclusion part. "In the future, gas extinction measurements at extended wavelengths (near-infrared) and for additional gases (e.g.,  $N_2$ ) will expand the spectroscopic applications in atmospheric studies." (Line 490-491)

Finally, the choice of colours hinders accessibility for many colourblind people. There are combinations of colours for use in figures that will make the results more readable for colourblind people. Printing in black and white will quickly show where the use of colours should be improved.

Reply: We updated the colors in the figures and used different symbols for different data sets.

Figure 1, data sets in panel a,b,c are mainly differentiated by marker shape. We kept the colors of the markers but enlarged the size of the markers. The colors in panel c and d are improved. We also use different markers for different data sets.

Figure 4, we used different markers for extinction coefficients at different wavelengths.

Figure 8, we changed the colors in panel c and used different symbols for those three data sets.

Figure 9, we updated the colors and used different symbols for the data points.

Figure S1 and Figure S4, we updated the colors of the traces and used different line styles.

Figure S3, we changed the colors of the markers and used different symbols.

**1. Specific remarks**

Line 43: please refer to the original year of publication in the reference (1899), not the year of the re-issue of the collected papers. Anyone familiar with the subject matter will be confused by Strutt 2009, but at the same time know immediately what Strutt 1899 is.

Reply: This reference was changed to the original one. (Line 43)

Strutt, J. W.: XXXIV. On the transmission of light through an atmosphere containing small particles in suspension, and on the origin of the blue of the sky, London, Edinburgh Dublin Philos. Mag. J. Sci., 47, 375-384, 10.1080/14786449908621276, 1899.

Line 50 (and several other locations throughout the manuscript): really weird line breaks.

Reply: These line breaks were fixed. (Line 51)

Line 74: Nitrogen should have subscript "2" not "e".

Reply: "Ne" was changed to "N2". (Line 74)

Line 99 in the Methods section: I found Bluvshtein et al., 2016 (doi:10.5194/amt-9-3477-2016) to be the end point of a chain of references for the method that starts with the two that are mentioned. I suggest to use this reference (in addition or instead of). Sending readers into a rabbit hole to chase the methods is not nice.

Reply: The BBCES system has two channels. The first publication describing the  $BBCES_{UV}$  channel (307-350 nm) is by Washenfelder et al., (2016), who used it to determine nitrogen dioxide and formaldehyde. Bluvshtein et al. (2016) used this system for aerosol light extinction measurements. These two papers were cited here, and we delete Bluvshtein et al. (2017). (Line 101)

Bluvshtein, N., Flores, J. M., Segev, L., and Rudich, Y.: A new approach for retrieving the UV–vis optical properties of ambient aerosols, Atmos. Meas. Tech., 9, 3477-3490, 10.5194/amt-9-3477-2016, 2016.

Washenfelder, R. A., Attwood, A. R., Flores, J. M., Zarzana, K. J., Rudich, Y., and Brown, S. S.: Broadband cavity-enhanced absorption spectroscopy in the ultraviolet spectral region for measurements of nitrogen dioxide and formaldehyde, Atmos. Meas. Tech., 9, 41-52, 10.5194/amt-9-41-2016, 2016.

Line 226: In the results and discussion I read: The reflectivity of the cavity mirrors, measured across the entire range using the difference in Rayleigh scattering of  $N_2$  and He, was very stable throughout the experiments". I expect to find this in the method section, how this was determined.

Reply: Reflectivity measurements were repeated every three samples measurements. The average peak reflectivity of the BBCESUV mirrors was 0.99933, with a 1 $\sigma$  uncertainty of 0.000006 at 330 nm. The average peak reflectivity of the BBCESVis mirrors was 0.9999550, with a 1 $\sigma$  uncertainty of 0.0000006 at 657.9 nm.

In method Section 3.1, we added "Reflectivity measurements were repeated every three samples measurements to track the stability of the system." in Line 139-140

In Section 3.1, we added the  $1\sigma$  uncertainty for the reflectivity. "The mean peak reflectivity of the BBCESUV mirrors was 0.999328±0.000006 (672±6 ppm) at 330 nm, with a corresponding effective optical pathlength of 1.40±0.01 km. The reflectivity curve of the BBCESVis is much more structured, with reflectivity ranging between 0.999224±0.000010 and 0.9999550±0.0000006 (45 776±10 ppm < loss < 776 45±0.6 ppm)..." Line 249-253

Line 231: the reflectivities and the losses have their relative order inter-changed, please maintain order for readability.

Reply: We changed "(45 ppm < loss < 776 ppm)" into "(45 776±10 ppm < loss < 776 45±0.6 ppm)" Line 252-253

Line 285: It is worthwhile to note what transitions of the  $O_2$ - $O_2$  absorption features occur here, and especially that both molecules leave the interaction in an electronically excited state, at least for the shorter wavelengths.

Reply: The transitions of the O2-O2 absorption features around 688, 629, and 580 nm are  ${}^{1}\Sigma_{g}^{+}(\nu = 1)$ ,  ${}^{1}\Delta_{g} + {}^{1}\Delta_{g}$  ( $\nu = 0$ ), and  ${}^{1}\Delta_{g} + {}^{1}\Delta_{g}$  ( $\nu = 1$ ), respectively. This is added in Line \*\* as "...overlap with O2-O2 CIA bands of are  ${}^{1}\Sigma_{g}^{+}(\nu = 1)$ ,  ${}^{1}\Delta_{g} + {}^{1}\Delta_{g}$  ( $\nu = 0$ ), and  ${}^{1}\Delta_{g} + {}^{1}\Delta_{g}$  ( $\nu = 1$ ), respectively. These absorption bands..." (Line 306-307)

**2. Figures and captions**

Line 466: Caption does not match labels in Figure. Given the magnitude of the signal in the figure, the error is in the caption, not the figure.

Reply: The caption is now revised as "Figure 1. Rayleigh scattering cross-sections of  $CO_2$  (a),  $SF_6$  (b), and N2O (c). Panel (d) shows the relative standard deviations..."

Figure 3: missing units on the axes.

Reply: We put the unit in the figure caption "Figure 3. Correlations between the extinction coefficients (unit,  $cm^{-1}$ ) measured by the BBCES and CRDS."

Figure 4: consider listing tabular material in a table rather than a figure legend.

Reply: We listed the fitted coefficients in a table next to the figure.

Figure 5: the unit of panel (c) cannot be correct, there must be a density involved here.

Reply: The unit is now revised as "cm5 molecules-2".

Figure 6: the methane percentage is mentioned, but not the cell density for 100% methane concentration.

Reply: We added the cell density for 100% methane concentration in the caption as "The number concentration of 100% methane was  $2.50143 \times 10^{19}$  molecules cm-3."

Figure 6 & 7: suggest to add markers to figure 6 indicating where the wavelengths are that are shown in figure 7.

Reply: Thank you for this suggestion. We added vertical lines to indicate the wavelengths that are shown in Figure 7. We also illustrated in the figure caption as "The selected wavelengths are shown in Figure 6 by vertical lines."

Figure S3: missing units on the axes

Reply: We provide the unit in the figure caption.

Figure S4: in black white the traces look identical.

Reply: We changed the colors and line style of the traces.

---

## Author Comment (AC1) · 4 Apr 2021

The comment was uploaded in the form of a supplement:
https://acp.copernicus.org/preprints/acp-2020-941/acp-2020-941-AC1-supplement.pdf

---

## Author Response (AR1)

The authors report results from a laboratory study using Broadband Cavity Enhanced Spectroscopy (BBCES), supplemented by Cavity Ring Down Spectroscopy (CRDS) to measure Rayleigh scattering cross sections and absorption cross sections, where applicable, for the gases $CO_2$, $N_2O$, $SF_6$, $O_2$, and $CH_4$. What's new here is the use of a single mirror to cover a very broad wavelength range for the BBCES studies from 338-725 nm. The topics covered are important and within the scope of ACP, but I have significant concerns that need to be addressed before this paper can potentially be published.

Reply: We thank the Reviewer for the insightfull comments and the careful reading of our manuscript. We appreciate the comments that substantially helped to improve the manuscript.

— Major Comments/Corrections:

(1) The paper makes claims of superiority over prior work that do not appear to be supported by the data. For example, this sentence in the Abstract that is echoed in various other forms in the paper: "This study provides improved refractive index dispersion relations, n-based Rayleigh scattering cross-sections, and absorption cross-sections for these gases." In a few instances, the data from this paper do represent clear improvements over what is already in the literature (e.g., the significantly expanded wavelength range for the $N_2O$ Rayleigh scattering cross sections), but in many cases, the data from this paper can only be stated to be in agreement with past work within the stated uncertainties rather than "improved". The authors should carefully go through the manuscript and make sure they are not overstating the superiority of the present

results over past studies.

Reply: We appreciate the reviewer's concern. As the reviewer suggests, the characterization of an improved dispersion relation is based on the extended wavelength range and continuous nature of the spectra compared to the literature. We have carefully gone through the manuscript to ensure that the language accurately represents cases in which our data agree with the literature and those in which is provides additional measurements. We have rephrased the sentence in the abstract to read "This study provides dispersion relations for refractive indices, n-based Rayleigh scattering cross sections and absorption cross sections based on more continuous and more extended wavelength ranges than available in the current literature.

This concern is especially relevant for the revised dispersion relations for SF6 and CH4. Based on the large spread in the n values from this study (and the range of pink shading, which I assume is the 1-sigma uncertainty) in Figure 9a and 9c, it seems clear that the data from this study do not provide constraint on the dispersion relations to allow distinguishing between the black and green fit lines. Moreover, the Shardanand and Rao data, which have large uncertainties and were intentionally excluded from analyses in earlier studies, only serve to increase the uncertainty of the CH4 fit here rather than decrease it. There are other issues with Figure 9 to be addressed later, but the point here is that the inclusion of the pink points from this study and black dots from Shardanand and Rao do not provide improved dispersion relations for SF6 and CH4 over the existing n-based relations. I find it disconcerting that nearly all of the pink points at wavelengths shorter than 480 nm lie above the fit line for SF6. Similarly, for CH4 nearly all of the pink points shorter than 400 nm are above the fit line except in the region 340–360 nm, where all points lie below the fit line. There appear to be systematic biases as a function of wavelength. Moreover, there is no discussion in Section 3.5 of how the different data sets were weighted to produce the fits.

Reply: We realized that these concerns originate from the fact that the refractive index data were not scaled to the same condition of 288.15K and 1013.25 hPa. The problem was also pointed out by the Reviewer in the later part of the comments. We have

therefore scaled all the data (ours and literature) in the revised manuscript to these conditions to enable better compasrison. After scaling, our data were brought closer to the literature data. See the revised Figure 9 below.

In the revised manuscript, the Shardanand and Rao data, which have large uncertainties, were excluded during the fitting. After adding more data from our study, we weighted all of the data sets equally for fitting, which is different from the approach of Wilmouth and Sayres (2020), where the data sets were differently weighted during the fitting. Our data now lies around the fitted lines and in most cases, it falls well within 1-sigma uncertainty (pink shaded) of our data. Thus, we believe that our data adds constraints for the fitting.

The calculated Rayleigh scattering cross sections using the dispersion relations derived in this study were compared with those derived from previously recommended formulations listed in Table 1 (Figure 9). The difference increases towrds the longer wavelength region between 320 and 725 nm (Figure S4). The average deviations are 0.1%, 0.9%, and 0.1% for $SF_6$, $N_2O$, and $CH_4$, respectively. The derived dispersion relationships agree very well with those of Wilmouth and Sayres (2020), as shown in Figure 9 (c-d).

Thus, we revised section 3.5 due to changes in the data processing. (Line 539-617)

The scaling method of the refractive index can be found in the Methods part "Additionally, the refractive indices of $SF_6$, $N_2O$, and $CH_4$ were calculated based on Equation (1) using cross section results from this study and the King correction factors listed in Table 1. Our measurements were performed under ~295K and ~1020 hPa. However, the calculated refractive indices were scaled to 288.15K and 1013.25 hPa as in previous studies (Sneep and Ubachs, 2005; Wilmouth and Sayres, 2020)."(Line 262-266).

The data weighting during the fitting is described as "For the formulation of the refractive index of $CH_4$, Wilmouth and Sayres (2020) weighted the data sets from Watson and Ramaswamy (1936) and Cuthbertson and Cuthbertson (1920) equally but gave more weight to their UV measurements when deriving the formulation of the refractive index. In this study, all of the $CH_4$ data set were weighted equally. The derived

dispersion relation agrees very well with that from Wilmouth and Sayres (2020), as shown in Figure 9 (c-d)." (Line 595-617)

[Figure]

Figure 9. Real refractive index (*n*) for SF$_6$ (a), N$_2$O (b), and CH$_4$ (c). Comparison of Refractive index from this work with previous studies (Cuthbertson and Cuthbertson, 1920; Naus and Ubachs, 2000; Shardanand and Rao, 1977; Sneep and Ubachs, 2005; Vukovic et al., 1996; Watson et al., 1936; Wilmouth and Sayres, 2019, 2020) over the wavelength range of 264–725 nm. The green line represents the dispersion relation given in Table 1. The black line represents the dispersion relation given in Eq. (8–10) derived from a fit to our data and references results. The shading represents 1-σ uncertainty of *n*. The *n* values for Shardanand and Rao (1977), Sneep and Ubachs (2005), Naus and Ubachs (2000) were calculated from their reported Rayleigh scattering cross sections. Refractive index data from Sneep and Ubachs (2005) are not used in the fitting since these results are away from others. Data from Shardanand and Rao (1977) are not used due to large uncertainties. All of the data sets were equally weighted during fitting. Panel (d) is a close-up view of panel (c) in the wavelength range of 264–363 nm.

In short, I do not see anything convincing in the manuscript that the dispersion relationships produced for SF6 and CH4 are "improved" relative to the latest n-based

values in the literature, and it is very probable that the relations here are less accurate (see below). I recommend taking a consistent approach for SF6 and CH4 as to that used with CO2, which is to simply say that your results agree with the existing n-based values in the literature, rather than unnecessarily providing new dispersion relationships for SF6 and CH4 that cannot be distinguished from current values within your uncertainty.

Reply: As explained above, we updated the refractive index data for fitting, and all of the data sets are weighted equally during fitting. The derived dispersion relation agrees with that from Wilmouth and Sayres (2020), as shown in Figure 9 (c-d).

The calculated Rayleigh scattering cross sections using the dispersion relations derived in this study were compared with those derived from previously recommended formulations listed in Table 1 (Figure 9). The difference increases towards the longer wavelength in the region of 320–725 nm (Figure S4). The average deviations are 0.1%, 0.9%, and 0.1% for $SF_6$, $N_2O$, and $CH_4$, respectively. Our results based on data at an extended wavelength range support the reference results. Since our results agree well with results from Wilmouth and Sayres, we deleted the word "improved" in the abstract, and now state in the conclusion part that "The derived dispersion relations for $SF_6$ and $CH_4$ agree well with those provided by Wilmouth and Sayres (2020)." (Line 636-637)

(2) The accuracy of the Rayleigh scattering cross sections appears to be overstated. In section 2.4, the pressure is stated to be accurate to 0.01% and temperature to 0.1%. While the authors likely have the precision to measure changes in P and T to these levels, this level of accuracy would be surprising. Most pressure gauges are not even capable of 0.01% accuracy. How and where were P and T measured? How were the P and T gauges calibrated to this accuracy? Please provide part numbers for all devices used in the experiments.

Reply: The gas temperature was measured by a K-type thermocouple and the cavity pressure was recorded by a pressure gauge (Precision Pressure Transducer, Honeywell International Inc., MN, USA). The two BBCES channels were connected in series. These two parameters were measured between the two BBCES cavities. The pressure

gauge and the thermocouple were calibrated by the manufacturer. The temperature in our laboratory is very stable, and the daily variations are within ±0.25 °C. The system was operated at a low flow rate, and there is negligible resistance in the system. The stable environment in the lab helped to obtain temperature and pressure data with very small uncertainty. These details were provided in the manuscript as "The gas temperature (K-type thermocouple) and cavity pressure (Precision Pressure Transducer, Honeywell International Inc., MN, USA) were recorded between the two cavities for gas number density ($N$) calculation." (Line 233-235)

More importantly, the authors appear to be claiming that they have determined the reflectivity of the mirrors with essentially no experimental uncertainty. The 1.03% stated uncertainty in section 2.4 – which is the uncertainty in the effective pathlength, not the mirror reflectivity (please correct this misstatement in line 231) – is almost entirely due to the 1.0% N2 reference cross section. But what about the standard deviation of the N2/He runs? Surely, there was some spread in the data from run to run? And there undoubtedly was a wavelength dependence in the data quality based on mirror reflectivity, mirror transmission, fluctuations in the light source, the amount of light to the detector, etc.

Reply: In this study, the standard error from each measurement was used for the uncertainty calculation. As stated in the Methods part, 150 spectra were recorded for each gas measurement. The standard error of these 150 spectra was used to represent the uncertainty of the light intensity signal measurement. This uncertainly included reflects the variation in the light source, the fluctuations in the cavity and in the spectrometer. The relative uncertainty was ≪0.2%, as stated in the manuscript.

In the revised manuscript, we changed "mirror reflectivity" to "effective pathlength" (Line 313)

There is an additional problem here in that the reference N2 cross section in Table 1 is only valid up to 468 nm. At longer wavelengths, the 1% error should be increased to account for the uncertain extrapolation of this dispersion relation.

Reply: We agree with the Reviewer that the uncertainty of the Rayleigh scattering cross section for $N_2$ over 468 could be larger than 1%. However, no reference data is available to calculate how big it could be. Thus we added a note in the revised manuscript to explain this issue.

"The uncertainty for the Rayleigh scattering cross section of $N_2$ is validated up to 468 nm. The uncertainty above this wavelength may be larger than 1%, which is the value used for the calculation in our study. Thus, the uncertainty at wavelengths longer than 468 nm may be underestimated." (Line 321-338)

Most concerning of all with respect to stated uncertainties is that for the individual gases, the authors appear to be defining the 1-sigma uncertainty in their results based on the differences between their values and the existing n-based values. If this was indeed the approach taken, this is wrong and must be corrected. If this was not the approach taken, the method used should be clarified. Lines 262–264 simply state uncertainty values of 1.5%, 1.1%, and 1.5% for CO2, N2O, SF6, respectively, without explanation where they came from – these numbers appear again in the Conclusions Line 434 when these exact same numbers are stated to be the relative difference with the n-based calculations. As I examine the data plotted in Figure 1d, it is clear that the precision of the data in this study is generally quite good. But Figure 1e reveals that the accuracy is substantially worse than the precision. There is nearly +4% to − 4% discrepancy for CO2 and SF6 at some wavelengths relative to the n-based values. Whether or not the n-based values are correct is not the point here – the point is that the n-based values are smooth – so all of this chatter is due to the experimental data in the present paper. For CO2 and SF6, there appears to be a 7% swing just in going from around 590 to 600 nm. And this large error is in the mean values –the variation would be even greater in the individual runs. I suspect that the highly structured reflectivity curve (Figure S1) is creating problems, i.e., systematic biases creating persistent inaccuracies across the wavelength range even though the data reproducibility (precision) is good.

In short, for all the reasons mentioned above, the uncertainties should be completely revisited, more thoroughly explained, and almost certainly increased significantly from

the present values.

Reply: For the measurement of each gas, the light intensity, temperature, and pressure data were acquired 150 times. The standard error of these 150 single measurements was used as the 1-sigma uncertainty of each parameter. The same was done for the following helium measurement. Finally, the uncertainty of the effective pathlength, the uncertainties of the target gas measurement, and the uncertainties of the helium measurement were propagated together to calculate the 1-sigma uncertainty of the extinction cross sections. In the revised manuscript, we added these descriptions: "Each parameter (temperature, pressure, light intensity) was measured 150 times for each gas. The standard error of each parameter obtained from the 150 single measurements was used to calculate the uncertainty." (Line 308-310)

There was a mistake in the original manuscript (Line262-264) about the uncertainty of the $CO_2$, $N_2O$, and $SF_6$. This has been corrected in the revised manuscript as "The mean 1-σ uncertainty of the reported cross sections for all three gases across the 307–725 nm wavelength range is 1.04% for $CO_2$, 1.05% for $N_2O$, and 1.04% for $SF_6$." (Line 361-363)

We agree with the Reviewer that the large discrepancies around 590 to 600 nm are likely correlated with the structured mirror reflectivity. However, it is impossible to quantify how much this would add to the uncertainty of our data. To address this issue, several things were done to evaluate this.

1) First, the relative difference between our measured data and n-based calculations were calculated again and reported in the form of mean±SD. The mean number would give a measure of the bias relative to the n-based measurement and the standard deviation would give a measure of the error about this number. These numbers are now added to the revised manuscript: "The relative difference between our measurements and $n$-based values are (0.37±1.24)%, (–0.55±1.06)%, (0.91±1.35)% (Mean±SD) for $CO_2$, $N_2O$, and $SF_6$, respectively."(Line 391-393), "our extinction cross sections agree well with the $n$-based values with an average deviation of (2.81±1.21)%."(Line 475-476),"with an average difference of (0.89±2.18)%."(Line 531), "Comparison of our

measurements with the *n*-based calculations for these gases in the entire wavelength range of this study yields excellent agreement with relative differences of $(0.37\pm1.24)\%$, $(-0.55\pm1.06)\%$, $(0.91\pm1.35)\%$, $(2.81\pm1.21)\%$, and $(0.89\pm2.18)\%$, respectively."(Line 623-626).

2) To evaluate the uncertainty related to the structured mirror reflectivity, we fitted our Rayleigh scattering cross section data and the n-based data to a power function, $\sigma=A\lambda^B$. The difference between these two functions are now shown in Figure 1 panel (f). That would also be a meausure of the uncertainty comparing smooth functions to smooth functions. The change in the varability of the relative difference would represent the influence of the structured mirror reflectivity.We added the following text in the revised manuscript and add panel (f) in Figure 1.

"The relative difference between our measurements and the *n*-based values are $(0.37\pm1.24)\%$, $(-0.55\pm1.06)\%$, $(0.91\pm1.35)\%$ (Mean$\pm$stdev) for $CO_2$, $N_2O$, and $SF_6$, respectively. Variability of the relative difference is due to structure in the mirror reflectivity that does not fully cancel. The wavelength-dependent Rayleigh scattering cross section is generally described in the form of $\sigma = A \times \lambda^B$ In this study, the measured values and the n based data were both fitted to this function. The relative difference between these two fitted functions is shown in Figure 1(f). That would be a measure of the uncertainty comparing smooth functions to smooth functions. The relative differences were $(0.49\pm0.48)\%$, $(-0.41\pm0.30)\%$, and $(0.94\pm0.22)\%$ (Mean$\pm$stdev), for $CO_2$, $N_2O$, and $SF_6$, respectively. The mean values of the relative difference obtained from the fitting function are close to that obtained from the measurements. However, the variabilities are much smaller, which may be related to the cancellation of the influence by the structured mirror reflectivity.." (Line 391-402)

[Figure]

(3) The authors should cite previous studies where appropriate. Reference to the nbased values is made repeatedly throughout the paper for various gases, but it is never stated clearly from which study each of these n-based values originated. I recommend in column 5 of Table 1 that it be highlighted in some way (e.g., asterisk, bold, footnote) which of the listed References the n-based expression in column 2 came from. Also, the source of the n-based expression should be named for each gas at least once in the text and when comparisons are being made.

It would seem appropriate after Lines 80-81 to mention that Wilmouth and Sayres 2020 just published SF6 and CH4 dispersion relationships for the wavelength range 250-650 nm. Lines 90-93, 392-398, and 412-417 all discuss a method of determining new dispersion relationships for SF6 and CH4 in this study that is a copy of the Wilmouth and Sayres 2020 method, but the Wilmouth and Sayres 2020 paper is not credited for the idea that this paper is replicating.

Reply: We thank the Reviewer for this suggestion. In Table 1, we have labeled the source references of the refractive index (Bold) and the King correction factors (Italics) that are used for the $n$–based calculation. This is also explained in the footnote of Table 1. Moreover, these references were cited in the main text in the Methods part as "or the 307–725 nm wavelength range of this study, the $n$-based calculated Rayleigh scattering cross sections from largest to smallest are $SF_6$ (Sneep and Ubachs, 2005; Wilmouth and Sayres 2020), $N_2O$ (Sneep and Ubachs, 2005), $CO_2$ (Alms et al. 1975; Bideau-Mehu et al. 1973), $CH_4$ (Sneep and Ubachs, 2005; Wilmouth and Sayres 2020), $N_2$ (Bates 1984), $O_2$ (Bates 1984; Sneep and Ubachs, 2005), and He (Abjean et al., 1970; Cuthbertson and Cuthbertson, 1932)." (Line 248-262)

We also added the study by Wilmouth and Sayres in the revised introduction part as "Recently, Wilmouth and Sayres (2020) combined refractive index data in the UV region (264-297 nm and 333-363 nm) and at several single wavelengths in the visible, and they derived the dispersion relation of refractive index for $SF_6$ and $CH_4$ in the wavelength range of 264-650 nm. However, more data in the visible range are needed in order to further validate these dispersion relations." (Line 98-102)

In the revised manuscript, the method of determining the new dispersion relationships for $SF_6$ and $CH_4$ as described by Wilmouth and Sayres was added to the introduction. (Line 98-102) We also emphasized this in Section 3.5 where we derived the dispersion relation for $SF_6$ and $CH_4$. (Line 542-544)

(4) Figure 9 contains significant errors. The n-baqsed data in Figure 9 for the studies of Watson, Cuthbertson, and Vukovic are all plotted incorrectly, causing them to appear much further from the fit lines than they should be. The authors appear to have taken the index of refraction data directly from the papers and did not appropriately convert to the reference temperature of 288.15 K. The authors should also check Figure 8 for accuracy, where the same reference studies are plotted.

Reply: We thank the Reviewer for this comment. We have scaled the refractive index data from Watson and Ramaswamy 1936, Cuthbertson and Cuthbertson 1920, Vukovic et al. 1996 to 288.15 K and 1013.25 hPa. The cross sections were recalculated using the correct temperate and pressure. Figure 8 and Figure 9 were updated in the revised manuscript.

(5) Due to the errors highlighted in point (4) above, the dispersion relationships produced in this study that included the literature index of refraction data are therefore all incorrect. The authors also do not state how they converted their own Rayleigh scattering data to index of refraction, nor the temperature and pressure at which their dispersion equations were defined; these things should be addressed.

Again, once the data themselves are plotted correctly, I see no point in producing new dispersion formulas for SF6 and CH4 that cannot be distinguished from the existing

dispersion formulas within uncertainty. Simply state that the data here agree with existing values, as was done with CO2.

Reply: In the revised manuscript, we added more details about how we converted our scattering cross section data to the refractive index at 288.15K and 1013.25 hPa. "Additionally, the refractive indices of $SF_6$, $N_2O$, and $CH_4$ were calculated based on Equation (1) using cross section results from this study and the King correction factors listed in Table 1. Our measurements were performed under ~295K and ~1020 hPa. However, the calculated refractive indices were scaled to 288.15K and 1013.25 hPa as in previous studies (Sneep and Ubachs, 2005; Wilmouth and Sayres, 2020)." (Line 262-266)

After correction for all of the refractive index data, we derived the dispersion relations for again for $SF_6$, $N_2O$, and $CH_4$ and modified the results and discussion in Section 3.5 accordingly. (Line 539-617)

Wilouth and Sayres (2020) derived the dispersion relations of refractive index based on data in the UV (264-297 nm and 333-363 nm) and in the visible wavelength range. However, we note that only a limited number of data points in the visible wavelength range were used. For $SF_6$, there is only one data point at 633 nm and for $CH_4$, there are 12 points. In the present study, however, we obtained many more measurements in the visible wavelength range. Incorporating our results for the formulation of the refractive index is therfore beneficial. Thus, we consider it useful to derive the dispersion relation using our data. We found that the fitting results from our study agree well with the data from Wilmouth and Sayres (2020). We added this discussion in the revised manuscript. "The calculated Rayleigh scattering cross sections using the dispersion relations derived in this study were compared with those derived from previously recommended formulations listed in Table 1 (Figure 9). The difference increases towards the longer wavelength in the region of 320–725 nm (Figure S4). The average deviations are 0.1%, 0.9%, and 0.1% for $SF_6$, $N_2O$, and $CH_4$, respectively. Notably, the difference for $N_2O$ is more significant than for the other two gases. This study uses refractive index data in continuous wavelength ranges of 307–725 nm to derive the dispersion relation, while the formulation for $N_2O$ in Table 1 is derived by Sneep and Ubachs (2005) based on

polarizability measurements at five single wavelengths. For the formulation of the refractive index of $CH_4$, Wilmouth and Sayres (2020) weighted the data sets from Watson and Ramaswamy (1936) and Cuthbertson and Cuthbertson (1920) equally but gave more weight to their UV measurements when deriving the formulation of the refractive index. In this study, all the $CH_4$ data sets were weighted equally. The derived dispersion relation agrees very well with that from Wilmouth and Sayres (2020), as shown in Figure 9 (c-d)." (Line 588-617)

— Additional Comments/Corrections:

Line 1 and elsewhere: Cross-sections should not be hyphenated

Reply: We have changed "cross-sections" to "cross sections" in the revised manuscript.

Lines 60–62: It is typical to order references in a list like this chronologically. Add Thalman et al. 2017 (Erratum to 2014 paper) and Wilmouth and Sayres 2020 to this list.

Reply: We added these two references and updated the original reference. (Line 79)

Line 98 and many other places in the manuscript: The authors use "BBCES" as if it were the name of their experimental setup. If the sentence doesn't make sense with "Broadband Cavity Enhanced Spectroscopy" substituted in place of "BBCES", the sentence should be edited.

Reply: We thank the Reviewer for this suggestion. Line 98 using BBCES is correct because the whole paragraph is the introduction of BBCES. We went through the entire manuscript again. We made several changes in the manuscript: 1) Line 306, change "**2.4 Error Propagation for BBCES**" to "**2.4 Error Propagation for Extinction Measurements**", 2) Line 632, change "using BBCES" to "using BBCES and CRDS".

Lines 104-120: Please provide more detail on how the broad wavelength range was filtered and how the data acquisition worked. Did the spectrometer scan through the wavelength range X nm at a time? Were 3 seconds of data acquired at each wavelength per spectrum, or it took 3 seconds to scan the entire range? With light being present

over such a broad wavelength range, how was stray light prevented from impacting the results? It seems like this would have been a problem. For example, did you ever use a longpass filter to block the shorter wavelength light to see if there was still signal present at the detector at the shorter wavelengths due to stray light from the longer wavelengths?

Reply: There are two filters for each BBCES Channel. The UV channel has two filters from Schott Glass: The WG310 filter was used to filter light with wavelength shorter than 310 nm, and a bandpass filter UG11 which has high transmission between 260 and 400 nm. The visible channel also contains two filters: Schott WG 345 was used to filter light with wavelength <345 nm, while a short pass filter (Edmund Optics, 15-261) was used to filter light with wavelength >700 nm. This information was added in Line 136 and Line 140-141.

The spectrometer has a back-illuminated CCD array detector with 1024×56 pixels. This detector records data from 302.15−727.29 nm simultaneously. It does not scan from one wavelength to another. In our experiment, the CCD array was exposed for 3 s to obtain one spectrum. This information was already provided in the manuscript.

We used filters to remove light at both short and long wavelength range (short pass and long pass filters) to match the transmitted light wavelength range with the operating wavelength range of the high-reflectance mirrors. As indicated by the manufacturer, the stray light has a minimal influence on the spectra, e.g., stray light: <0.08% at 600 nm; 0.4% at 435 nm. Moreover, the test suggested by the Reviewer has, indeed, been carried out many times within different wavelength range when we searched for proper filters that fit the high reflection mirrors in the cavity. We did not detect any signal present at the detector at the shorter wavelengths due to stray light from the longer wavelengths.

Line 129: Why not present this equation already solved for R? Section 2: There is an awkward switching between past and present tense throughout. Pick one (ideally past) and stick with it.

Reply: The equation has been changed to show the solved R (Line 170). We went

through Section 2 and changed it to past tense.

Line 155 and many other places: The authors use "CRDS" as if it were the name of their experimental setup. If the sentence doesn't make sense with "Cavity Ring Down Spectroscopy" substituted in place of "CRDS", the sentence should be edited.

Reply: We have checked through the manuscript. The use of "CRDS" is correct.

Line 272: Delete "theoretical". The n-based calculations are based on experimentally determined data.

Reply: This has been corrected in the revised manuscript.

Line 345: Shouldn't 630 nm here be 629 nm, per Line 321?

Reply: Yes, it should be 629 nm. This was corrected.

Lines 346-348: Some of these discrepancies appear quite large, and the explanation given that it is because they are the lowest intensity bands is not entirely true. The band at 532 nm appears smaller than the band and 360 and 380 nm, and yet it agrees well. What the bands with larger discrepancies have in common is not their size, it's that they are all located at wavelengths below 450 nm, while the ones that agree well are all located at wavelengths above 450 nm. Please comment on why this wavelength dependent disagreement might be the case. The text says Thalman and Volkamer 2013 are being plotted, but the legend in the figure says Thalman et al. 2014. Also, why is there no comparison in the text with the most recent Karman 2019 study?

Reply: We thank the Reviewer for raising this question. First, the wavelength ranges where larger discrepancies (344, 360, 380, and 446 nm) appear have lower light intensity than those wavelengths with smaller discrepancies (477, 532, 577, and 630 nm). At a lower light intensity level, a small change in the light source could result in a large change in the determined extinction measurement. This could be one of the reasons why those larger discrepancies occurred. Additionally, the absorption at 344, 360, 380, and 446 nm contribute less to the total extinction as compared to that at 477,

532, 577, and 630 nm. For example, when the cavity is filled with 100% $O_2$, the absorption at 344, 360, 380, and 446 nm contribute 1.5%, 19.1%, 16.2%, and 12.8% of the total extinction, which are much smaller than those at 477 (64%), 532 (34%), 577 (88%), and 630 nm (90%). The low fraction of absorption may also induce larger discrepancies when apportionment absorption from total extinction. In the revised manuscript, we added more discussion on this: "Moreover, the absorptions at 344, 360, 380, and 446 nm contribute a much smaller fraction of the extinction as compared to that of 477, 532, 577, and 630 nm. Thus larger discrepancies were observed during the apportionment of absorption from extinction." (Line 489-491)

The paper Thalman and Volkamer 2013 should be used here. The Karman et al. 2019 paper explains the updates of the HITRAN collision-induced absorption section data. The $O_2$-$O_2$ collision-induced absorption data were taken from Thalman and Volkamer 2013. Thus we only compared the measurement results from Thalman and Volkamer, which is the original study.

Line 355-356: Please add a reference for this first sentence.

Reply: The paper Adel and Slipher, 1934 was added here. (Line 497)

Adel, A. and Slipher, V. M.: The Constitution of the Atmospheres of the Giant Planets, Physical Review, 46, 902-906, 10.1103/PhysRev.46.902, 1934.

Lines 371-373: This sentence is incorrect as written. The interferometer studies obtained index of refraction values, not scattering cross sections.

Reply: This sentence has been revised. We added "and the refractive index" before "of $CH_4$". (Line 516)

Line 383: It seems like an oversimplification to say that there is "good" agreement here. There are significant discrepancies at many wavelengths.

Reply: Limited data about absorption cross section of $CH_4$ is avauialble in the literature. In addition, the absorption cross section varies significantly with wavelength. There are large discrepancies between these datasets, and to the best of our knowledge, there is not a recommended data set. Thus we do not discuss the comparison to literature data.

Only a brief description is provided in the manuscript. We revised this sentence to "At most spectral ranges, our results are in better agreement with the results from previous studies by Giver, (1978) and Smith et al.(1990)." (Line 532-534)

Line 391: Delete "much".

Reply: Deleted.

Line 392: "Better constrain" relative to what?

Reply: We derived the dispersion relation expression of $SF_6$ over a wider wavelength range as compared to previous studies (Wilmouth and Sayres). The word "better" was deleted in the revised manuscript.

Line 412: Change 9b to 9c

Reply: Changed to 8b.

Lines 429-432, Lines 361-363, and Lines 327-332: The BBCES and CRDS agreement is certainly a positive result, but be careful not to overstate what this means (lines 329-331). It's just two wavelengths, and there are clearly significant wavelength-dependent errors in the BBCES data. Moreover, the locations of the CRDS points are all offset from both the fit lines and the BBCES data in Figure 9, suggesting that the CRDS results do not provide any additional constraint on the accuracy of the BBCES results even at these two wavelengths.

Reply: We thank the Reviewer for this comment. We agree that there are wavelength-dependent errors in the BBCES data. The agreement between CRDS and BBCES data at 404 and 662 nm does not guarantee the results at other wavelengths. We have modified this sentence to "This excellent agreement between the instruments further substantiates the BBCES measurements and suggests that the accuracy of the BBCES at these two wavelengths is better than estimated in the error propagation above, where the $N_2$ refractive index was the largest uncertainty." (Line 462-464)

We scaled the refractive index data to 288.15K and 1013.25 hPa. Now the results from

CRDS agree much better with the results from BBCES as shown in Figure 9.

Line 442-445: This point was made earlier, but to reiterate here, the claims of superiority from this study are not justified. The Wilmouth and Sayres 2020 fit of the UV region for CH4 that is plotted in Figures 9c and 9d was purposefully weighted such that it better represented the existing index of refraction data in the visible region at the slight expense of the UV fit, as described in that paper. A UV-only fit is also presented in that paper. The fact that the fit from the present manuscript is higher for CH4 simply means that this new fit is not representing the visible data as well as the Wilmouth and Sayres 2020 expression.

Reply: We have scaled our refractive index data and literature data in the revised manuscript. By incorporating our revised data into the data used by Wilmouth and Sayres (2020), we derived the dispersion relation for the refractive index of CH4. The Wilmouth and Sayres 2020 fit of the UV region for CH4 that is plotted in Figures 9c and 9d was purposefully weighted such that it better represented the existing index of refraction data in the visible region at the slight expense of the UV fit. In our fit, we added more data points in the wavelength range of 302-400 nm, and we weighted all of the data sets equally. Our fits agree well with that of Wilmouth and Sayres. The calculated Rayleigh scattering cross section from these two fits has a maximum difference of 0.15% and averaged 0.12%. This means that our data do constrain the formulation of the refractive index.

Figure 8: There are many errors in the figure legends. Wilmouth, Watson, and Cuthbertson are all two-author papers, not "et al." Watson is misspelled. Cuthbertson is misspelled. Add a space after Rao. Add a space after Sneep.

Reply: Thanks a lot for the careful reading. These errors have been corrected. The revised figure has been shown below.

[Figure]

Figure 9: It is never stated what the shading represents in this figure.

Reply: The shading represents the 1-σ uncertainty of the refractive index. This was added to the caption of Figure 9.

Table 1: The He equation is incorrect. There is a missing decimal in 1.8102.

Reply: Corrected.

— Minor Corrections:

Line 22: Delete "and" after 1.5%

Line 31: Delete "as" after either

Line 44: Add "a" after For

Line 53: Change "index" to indices

Line 151: Change "systems" to spectroscopy

Line 152: Change "-validation of" to validate

Line 156: Add "was" after thus

Line 209: Change "overlaps" to overlap

Line 216: Delete "the" after of

Line 235: Change "resulted uncertainty of" to resulting uncertainty in

Lines 306 and 313: Avoid apostrophe "instrument's" – instead say wavelength resolution of the instrument

Line 328: Change "slops" to slopes

Reply: We thank the Review for the careful reading. These have been corrected in the revised manuscript.

---

## Author Response (AR2)

The authors have addressed many of my comments since the first review, and the manuscript is definitely improved. However, I still have some significant concerns that need to be addressed before the paper can be published.

We thank the Reviewer for these comments and suggestions. We will address them in the replies below.

--Major Comments:

(1) My biggest concern is with the uncertainty. The highly structured reflectivity curve of the mirrors significantly limits the accuracy of the Rayleigh scattering cross sections in this study and therefore limits the conclusions that can be drawn from this work. The unfortunate reality is that the very thing that is newest about this paper (the very broad wavelength mirrors) is the thing that seems to cause the most problems. The limitations on accuracy due to the structured reflectivity is abundantly clear in Figure 9. The Pink data simply cannot be used to provide any constraint on distinguishing the green (previous work) and black (this study) fit lines, especially for SF6 and CH4. Perhaps the green and black fit lines are different enough for N2O to warrant producing a new dispersion relation. But for SF6 and CH4, the results of this study can only be used to say that they validate the existing n-based expressions.

Reply: We agree with the reviewer that the structure in the derived refractive indices that arises from the residual structure in the reflectivity curves is a limitation. However, we respectfully disagree that the data are therefore of no value to the literature, as implied by the statement that we could not distinguish our own fit from the prior data. That these two fits are indeed very close together, and both well within the error bounds estimated for the BBCES data, speaks to the accuracy of the BBCES data and the prior determinations. The agreement between the CRDS data and the BBCES data at 405 nm and 662 nm is 0.24% and 0.8%, both values well within the stated uncertainty of the BBCES data in the figure. CRDS is an absolute method that does not suffer from the apparent issue of the structured mirror reflectivity. Moreover, the large number of data points in the BBCES data easily compensates for the uncertainty in the structured mirror reflectivity when fitting a smooth function to the data. For the case of SF6, for example, there are 327 points in the BBCES data and 2 additional CRDS points. The total number of literature data points across this wavelength range is 8 prior to the recent data from Wilmouth, and 57 including these recent data. The sparseness of the literature does not allow one to assess if there may have been systematic errors in these measurements similar to the structured mirror reflectivity artifact in the BBCES data, so any potential artifacts of this nature in the literature data are simply unknown. The improvement in the refractive index fit on going from 57 literature data points to 57+329 = 386 is a factor of 2.5.

To illustrate the importance of the denser wavelength coverage, we present in the figure below the same refractive index fit using our raw BBCES data and a series of fits to more highly averaged data. As the averaging increases, the apparent structure in the data decreases, but the fit does not change. We have added these figures as a supplement to the paper in the discussion surrounding Figure 9. We also illustrate this in the revised manuscript at the end of Section 3.5.

"The structure in derived refractive indices that arises from the residual structure in the reflectivity curves is a limitation. However, the agreement between the CRDS data and the BBCES data at 405 nm and 662 nm is 0.24% and 0.8%, both values well within the stated uncertainty of the BBCES data. CRDS is an absolute method that does not suffer from the apparent artifact due to the structured mirror reflectivity. Moreover, the large number of data points in the BBCES data easily compensates for the uncertainty in the structured mirror reflectivity when fitting a smooth function to the data. Figure S5–7 show the refractive index fit using our raw BBCES data and a series of fits to more highly averaged data. As the averaging increases, the apparent structure in the data decreases, but the fit does not change." (Line 484-492)

1.SF$_6$

[Figure]

Table 1. The difference between the fit and n-based calculations

| Types | Mean | Min | Max | STD |
|---|---|---|---|---|
| Raw | -5.28E-07 | -2.26E-06 | 3.68E-06 | 1.57E-06 |
| Ave_2.5 | -5.11E-07 | -2.17E-06 | 3.50E-06 | 1.50E-06 |
| Ave_5 | -5.37E-07 | -2.26E-06 | 3.59E-06 | 1.58E-06 |
| Ave_10 | -5.30E-07 | -2.26E-06 | 3.52E-06 | 1.60E-06 |
| Ave-20 | -3.07E-07 | -1.92E-06 | 3.22E-06 | 1.51E-06 |

2.N$_2$O

[Figure]

Table 2. The difference between the fit and n-based calculations

| Types | Mean | Min | Max | STD |
|---|---|---|---|---|
| Raw | -2.30E-06 | -2.87E-06 | -9.79E-07 | 5.05E-07 |
| Ave_2.5 | -2.30E-06 | -2.88E-06 | -9.82E-07 | 5.09E-07 |
| Ave_5 | -2.32E-06 | -2.91E-06 | -9.76E-07 | 5.28E-07 |
| Ave_10 | -2.30E-06 | -2.91E-06 | -9.47E-07 | 5.50E-07 |
| Ave-20 | -2.26E-06 | -2.81E-06 | -1.09E-06 | 5.10E-07 |

3. $CH_4$

[Figure]

Table 3. The difference between the fit and n-based calculations

| Types | Mean | Min | Max | STD |
|---|---|---|---|---|
| Raw | 5.81E-07 | 2.87E-07 | 7.97E-07 | 1.47E-07 |
| Ave_2.5 | 5.87E-07 | 4.32E-07 | 7.00E-07 | 1.40E-07 |
| Ave_5 | 6.06E-07 | 3.22E-07 | 8.20E-07 | 1.54E-07 |
| Ave_10 | 6.03E-07 | 3.94E-07 | 7.63E-07 | 1.24E-07 |
| Ave-20 | 6.03E-07 | 5.90E-07 | 7.94E-07 | 8.14E-08 |

Moreover, as now detailed in lines 254-259 of the revised manuscript, there are additional uncertainties beyond those treated in the paper that the authors cannot quantify and did not even attempt to estimate. Because the uncertainty of the N2 reference cross section is unknown at wavelengths greater than 468 nm and because the mirror reflectivity is highly structured, the authors do not know their true uncertainties, and

the values in the paper represent a lower limit. Indeed, because N2 is used to determine the pathlength, all of the Rayleigh scattering cross sections reported in this study have unknown uncertainties >468 nm.

Reply: We respectfully disagree. Naus and Ubachs (2000) conducted CRD measurement of Rayleigh scattering cross section [$\sigma(v)$] for $N_2$ in the wavelength range of 560-650 nm. They also calculated the Rayleigh scattering cross section for $N_2$ based on the refractive index calculated from the dispersion relation developed by Peck and Khanna (1966) [$(n-1) \times 10^8 = 6498.2 + \frac{307.4335 \times 10^{13}}{14.4 \times 10^9 - v^2}, \lambda = \frac{1}{v} > 468\ nm$]. The Rayleigh scattering cross sections were fitted to a function of $\sigma(v) = \bar{\sigma} \times v^{4+\varepsilon}$, where $\varepsilon$ is 0.0624 for $N_2$. The values for $\bar{\sigma}_{exp}$, obtained after several (weighted) fitting and data-analyzing procedures, allow for a comparison between the observed ($\bar{\sigma}_{exp}$) and the calculated values ($\bar{\sigma}_{cal}$) of the Rayleigh scattering cross section. They found that values for $\bar{\sigma}_{exp}$ is deduced with accuracies of ~1%. The cross sections calculated based on dispersion relation by Peck and Khanna (1966) and the dispersion relation used by this study agree with each other with an average difference of 0.08% (0% to 0.12%) in the wavelength range of 468-725 nm. Thus, the uncertainty of the n-based Rayleigh scattering cross sections used by this study is about 1%. Moreover, recent CRD measurements by Thalman et al. (2014) agree well with calculations based on the dispersion relation used by this study with relative differences within 0.5% (See the table below). All of this supports that the uncertainty of the Rayleigh scattering cross section of $N_2$ is about 1%. We add the following sentences in the manuscript as "Rayleigh scattering cross section of $N_2$ is validated up to 468 nm with an uncertainty of 1%. Rayleigh scattering cross sections measurement by CRDs agree well with those calculated from the refractive index with relative difference within 1% in the wavelength range of 468–650 nm (Naus and Ubachs, 2000; Thalman et al., 2014). Thus, an uncertainty of 1% was assigned for $N_2$ Rayleigh scattering cross section in the wavelength range of 307–725 nm." (Line 245-250)

We further point out that the BBCES measurements at 662 nm agree with the absolute CRDS measurements from this data set to within 0.9%. This agreement between the two measurements, one relative to the Rayleigh cross section of $N_2$ and the other absolute, demonstrates the accuracy of the BBCES data beyond 468 nm.

Rayleigh scattering cross sections (cm$^{-3}$ ×10$^{27}$) of $N_2$ from *n*-based calculations and CRDs measurements for selected wavelengths.

| Wavelength | n-based | CRD measured | Relative difference (%) | Reference |
|---|---|---|---|---|
| 405.8 | 16.146 | 16.1 | -0.3 | Thalman et al. 2014 |
| 532.2 | 5.297 | 5.32 | 0.4 | Thalman et al. 2014 |
| 404 | 16.45 | 16.42 | -0.2 | This lab |
| 662 | 2.181 | 2.18 | 0.0 | This lab |

Naus Hans, Ubachs Win. Experimental verification of Rayleigh scattering cross sections. Opt. Lett. 2000; 25(5): 347–9.
Peck Edson R., Khanna Baij Nath. Dispersion of Nitrogen. J. Opt. Soc. Am., 1966; 56:1059–63.

For these reasons, I would state even more strongly than in my previous review that there is no point in publishing new dispersion relationships in this paper that are indistinguishable within uncertainty from the dispersion relationships already in the literature. The authors take this approach for CO2, opting not to report a new dispersion relationship, and they should take a consistent approach with SF6 and CH4, given

that the uncertainties are too large to say anything other than the SF6 and CH4 measurements in this study agree with the dispersion relationships currently in the literature.

Reply: Once again, we respectfully disagree. We are in fact somewhat confused by this recommendation. Indeed, our data and analysis agrees with what is already in the literature. This is not a reason not to report it. Quite to the contrary, it is common practice to publish measurements of the same quantity made by different approaches to test the understanding of basic physical and chemical properties (see, for example, the kinetics literature (JPL and IUPAC panels)). Furthermore, our data set has greater spectral density and breadth of wavelength coverage than any currently available data set or in fact the sum of all of the literature reported to date. To simply not report a valid result arising from such a data set taken by a new and different technique would be negligent. This statement is valid regardless of the degree of agreement or lack thereof with the existing literature.

2) On Line 457, it is stated that the average deviations in the dispersion relationships between the literature and this study for SF6 and CH4 are 0.1% for both gases. On first read, this level of agreement would seem to be almost impossibly close given the large uncertainties in the present data set. But upon closer inspection, this remarkable agreement in reality is simply due to the fact that the authors are just repeating the method of Wilmouth and Sayres (2020) for determining the dispersion relations and adding in their data from this study. Therefore, the 0.1% agreement essentially just means that one gets the same dispersion relation using the Wilmouth and Sayres (2020) method for SF6 and CH4 regardless of whether or not the data from the present study is included in the fit. Again, there is no justification here for providing new dispersion relationships for SF6 and CH4.

Reply: We have added a sentence clarifying how this was done in the revised manuscript as "The fit performed by this study combines the data set used by Wilmouth and Sayres and the data acquired by this study. These small deviations support that our data acquired by the BBCES at a wide range agrees well with literature data." (Line 472-474)

(3) The authors have a new approach from their previous manuscript of presenting the differences between their Rayleigh scattering cross sections and the n-based values as x +/- y%, as in lines 22-23 of the Abstract and in several places in the text. For example, for CO2, the difference is listed at 0.37 +/- 1.24%. Some explicit discussion when this approach is first presented (around line 306) regarding why the standard deviation is so much larger than the mean would be helpful. There are also too many significant figures being used in this approach. I recommend at most 2 significant figures rather than 3, e.g., with CO2, it should be listed at 0.4 +/- 1.2%.

Reply: Thank you for this suggestion. We agree that it is reasonable to keep one figure after decimal point. We changed this in the revised manuscript. (Line 22-23, 257, 260, 294, 295, 317, 324, 386, 426, 500, 501)

The relative difference (RD) was calculated as RD = (measured value-n-based calculation)/n-based calculation. As we can see from the figures, the measured cross section can be bigger or smaller than the n-based values. Thus, we have positive and negative values for the RD. This causes the mean of the RD to be smaller than the standard deviation. Statistically, there is no direct correlation between the mean and the standard deviation value.

(4) Based on lines 241-247, the authors appear to be using their precision as if it were their accuracy. The variation in the temperature and pressure over the 150 scans is the precision, not the accuracy. This new

text now makes it clear why the pressure uncertainty is listed at the unrealistic value of +/-0.01%, as I noted in the previous review. I would be extremely surprised if the pressure gauge used in this study were capable of 0.01% accuracy. Please update the uncertainties with reasonable estimates for accuracy, not using the precision.

Reply: We re-calculated the uncertainty of our data. That is, we calculated the overall uncertainty as the root sum squares of the standard deviation of the measurements, and the uncertainties in the effective pathlength, the temperature, the pressure, and the photon counting uncertainty in the spectra. We revised Section 2.4 as follows:

"The uncertainty for BBCES measurements can be assessed by the propagation of the errors associated with the measurements. Rayleigh scattering cross section of $N_2$ is validated up to 468 nm with an uncertainty of 1%. Rayleigh scattering cross sections measurement by CRDs agree well with those calculated from the refractive index with relative difference within 1% in the wavelength range of 468–650 nm (Naus and Ubachs, 2000; Thalman et al., 2014). Thus, an uncertainty of 1% was assigned for $N_2$ Rayleigh scattering cross section in the wavelength range of 307–725 nm. Each parameter (temperature, pressure, light intensity) was measured 150 times for each gas. The standard deviation of the measurements (<0.3%) is combined with the uncertainties in the pressure (±0.3%), temperature (±0.1%), the Rayleigh cross section uncertainties for $N_2$ (±1%) as well as uncertainty in the measurements of the spectral signal by the spectrometer (≪0.2%) to get an overall relative uncertainty for the effective pathlength curve of ±1.1%. This uncertainty is further propagated to the target gas by consideration of the uncertainties of pressure (±0.3%), temperature (±0.1%), and spectral intensity (≪0.2%) of the target gas measurements, and the standard deviation of the measurements (<1.2%). The overall 1-σ uncertainty of the gas extinction cross section is within 1.7%." (Line 245-260)

We also updated the figures accordingly and relevant text in the manuscript.

(5) As stated in (2) above, the dispersion relationships for SF6 and CH4 in this study appear to have been calculated using the exact same method and data sets as Wilmouth and Sayres (2020) except that the data from the present study were included in the fit, and "all sets of data were weighted equally" according to the text. This latter point is stated as if it were important, but it is never actually explained in the text what weighting the data sets equally means.

Does weighting the data sets equally mean that the points from all of the data sets were combined, and then the fit obtained? If so, then the data set with the most points will have far more weight than the ones with fewer points, such that they will not be truly weighted equally. More specifically, with this approach, the Vukovic, Watson, and Cuthbertson studies will be mostly irrelevant in the fits because the BBCES studies have so many more data points.

Alternatively, weighting them equally could mean that each data set was normalized according to how many points were in the data set, such that each study got equal weight. The downside of this approach is that the older Watson and Cuthbertson studies are probably less accurate, but they would be given equal weight with the newer studies.

Either way, it should be clearly stated what was done. Or alternatively, as strongly recommended above, delete the dispersion relationship calculations altogether for the reasons outlined in points (1) and (2). Reply: During the fitting, we weighted each data point equally instead of weighting each data set equally. More specifically, the points from all of the data sets were combined, and then the fit was obtained. We agree with the reviewer that Vukovic, Watson, and Cuthbertson studies will weigh less in the fitting due to

fewer data points from these studies. Since the BBCES measurements cover a much wider wavelength range, these results which contain more data points should have heavier weight during the fitting. We revised this in the manuscript as "All data points were weighted equally." (Line 445)

--Additional Comments/Corrections:

Line 72 and many other Lines in the manuscript: I mentioned this in my previous review, but the problem persists. The authors use "BBCES" as if it were the name of their experimental setup. If the sentence doesn't make sense with "Broadband Cavity Enhanced Spectroscopy" substituted in place of "BBCES", the sentence should be edited. In general, anywhere in the paper that the expression "the BBCES" or "our BBCES" is written, it is incorrect. For example, "the BBCES setup" or "our BBCES instrument" could be stated instead.

Reply: We thank the reviewer for this suggestion. We agree that the results are obtained from the "BBCES system" instead of only "BBCES". The same case for CRD. We changed accordingly throughout the revised manuscript.

Lines 282-284. These values are not the "1-sigma uncertainty" as stated in the text. They appear to be from a root sum of squares calculation that includes the N2 reference cross section with the precision of the temperature, precision of the pressure, and precision of the test gas spectra. The true 1-sigma uncertainty is much large than the numbers presented in this sentence.

Reply: In the revised manuscript, the 1-sigma uncertainty was calculated as the root sum square of the uncertainties for the effective pathlength (1.1%), the temperature (0.3%), the pressure (0.1%), the photon counting of the spectra (0.2%) and the standard deviation of the measurements (<1.2%). We updated the text and figures accordingly. (Line 257-260, 294-295, Figures 1,5,8,9)

Line 429 and Line 438: Saying "for 288.15K and 1013.25 hPa" is ambiguous as written in these sentences, as it sounds like the measurements were made at these conditions. These T and P values should appear a bit later with the dispersion relationships to be clear that n is being defined at 288.15K and 1013.25 hPa in order to be consistent with past studies.

Reply: To make this sentence accurate, we changed it to "In our study, the refractive index of $SF_6$ in the wavelength range of 307-725 nm was calculated from the measured Rayleigh scattering cross section. The calculated refractive index was scaled to 288.15 K and 1013.25 hPa to be consistent with past studies." (Line 440-442)

Line 430: Delete "better" and Line 431: Delete "alternative". This is the standard form of the fit; it is not better or alternative to what has been done before.

Reply: These two words were deleted in the revised manuscript. (Line 442, 444)

Line 447: To avoid ambiguity, replace "their refractive index" with "Sneep and Ubachs (2005)".

Reply: Replaced. (Line 460)

Line 475: Please repeat the list of gases in order before "respectively". As it is, this sentence is referencing a gas list from two sentences prior.

Reply: Added. (Line 501-502)

Table 1: The He equation is still incorrect. There is a missing decimal in 1.8102.

Reply: Thanks for the careful reading. This has been revised.

Figure 5b: Missing right parenthesis in figure caption

Reply: We added the right parenthesis in panel (b)

Figure 9, panel (c): Is the green fit line plotted for CH4? It is not visible on the plot.

Reply: It is plotted. However, the green fit line overlaps with the black fit line.

--Minor Corrections:

Line 80: Change "is" to "are"

Reply: Changed. (Line 82)

Line 84: Add "applicable" after "CH4" and change "264" to "250"

Reply: Changed. (Line 86-87)

Lines 135-136: Delete the sentence beginning "In this study…" as it simply repeats what was just stated on Lines 126-127.

Reply: This sentence was deleted in the revised manuscript. (Line 139-140)

Lines 273-275: The sentence beginning "The mean uncertainty…" could be deleted as it repeats what was already stated on Lines 246-247.

Reply: This sentence was deleted in the revised manuscript. (Line 283-286)

Line 292: Change "listed" to "lists"

Reply: Changed. (Line 303)

Line 299: Add ", 2020" after "2019"

Reply: Added. (Line 310)

Line 301: Change "calculation" to "calculations"

Reply: Changed. (Line 312)

Line 309: Define variables in the equation

Reply: The sentence has been changed to "…described in the form of $\sigma = A \times \lambda^B$, where $\sigma$ and $\lambda$ are the cross section and the wavelength.". (Line 320-321)

Line 310: Change "fitted" to "fit"

Reply: Changed. (Line 321)

Line 359: Change "slops" to slopes

Reply: Changed. (Line 371)

--Note
A few of the changes indicated in the authors' response to the previous review were never actually made in the revised manuscript.

Reply: We thank the reviewer for point out this. We checked the manuscript and updated the manuscript.

1. In the abstract, we changed the last sentence to "This study provides dispersion relations for refractive indices, n-based Rayleigh scattering cross sections and absorption cross sections based on more continuous and more extended wavelength ranges that available in the current literature." (Line 26-29)
2. Change 630 nm to 629 nm. (Line 388)
3. Change "Figure 9b" to "Figure 9c". (Line 465)

--Final Suggestion

If someone were to summarize what's new about this paper in a single statement, it would be that the measurements were made with such broad mirrors. I think some perspective text at the end of the paper discussing pros and cons of using mirrors like these would be appropriate. It seems clear that there is a tradeoff one must make here between speed (acquiring a broad wavelength range quickly) and accuracy (the structured reflectivity curve is limiting).

Reply: We agree with this suggestion. While the BBCES instrument provides data at a wide wavelength range, the structured mirror reflectivity also induces some uncertainties to the wavelength ranges where the reflectivity changes significantly. We add this in the revised manuscript as "We also note that while acquiring data at a broad wavelength range quickly using a single BBCES instrument, uncertainties were also observed in the wavelength ranges where the mirror reflectivity changed significantly. However, with appropriate averaging, this can be minimized without compromising the accuracy. There is a tradeoff between obtaining data at a wide wavelength range and ensuring high accuracy data. New mirrors with a smoother reflectivity curve will improve the performance of BBCES instrument." (Line 540-546)